# Synergy of the westerly winds and monsoons in lake evolution of global closed basins since the Last Glacial Maximum and its implication for hydrological change in Central Asia

Yu Li[1], Yuxin Zhang[1]

[1]Key Laboratory of Western China's Environmental Systems (Ministry of Education), College of Earth and Environmental Sciences, Center for Hydrologic Cycle and Water Resources in Arid Region, Lanzhou University, Lanzhou 730000, China

*Correspondence to:* Yu Li (liyu@lzu.edu.cn)

**Abstract.** Monsoon system and westerly circulation, to which climate change responds differently, are two important components of global atmospheric circulation, interacting with each other in the mid-to-low latitudes. Relevant researches on global millennial scale climate change in monsoon and westerlies regions are mostly devoted to multi-proxy analyses of lakes, stalagmites, ice cores, marine and eolian sediments. Different responses from these proxies to long-term environmental change make understanding climate change pattern in monsoon and westerlies regions difficult. Accordingly, we disaggregated global closed basins into areas governed by monsoon and westerly winds, and unified paleoclimate indicators, as well as added the lake models and paleoclimate simulations for emphatically tracking millennial scale evolution characteristics and mechanisms of East Asian summer monsoon and westerly winds since the Last Glacial Maximum (LGM). Our results conclude that millennial scale water balance change exhibits an obvious boundary between global monsoon and westerlies regions in closed basins, particularly in the Northern Hemisphere. The effective moisture in most closed basins of the mid-latitudes Northern Hemisphere mainly exhibits a decreasing trend since the LGM, while of the low-latitudes shows an increasing trend. In the monsoon dominated closed basins of Asia, humid climate prevails in the early-to-mid Holocene and relative dry climate appears in the LGM and late Holocene. While in the westerly winds dominated closed basins of Asia, climate is characterized by humid LGM and mid-Holocene (MH) compared with the dry early and late Holocene, which is likely to be connected with precipitation brought by the westerly circulation. This study provides an insight into long-term evolution and synergy of westerly winds and monsoon systems and a basis for projection of future hydrological balance.

## 1 Introduction

As important components of atmospheric circulation systems, the mid-latitude westerly winds and low-latitude monsoon systems play key roles in global climate change. Whether on the decadal or the millennial scale, researches about this aspect always attract widespread attention from scientists. Examination of global monsoon precipitation changes in land suggests an overall weakening over the recent half-century (1950-2000) (Zhou et al., 2008). Individual monsoon indexes reconstructed

by Wang et al. (2017) indicate the moisture in the tropical Australian, the East Africa, and the Indian monsoon regions exhibits a gradual decrease since the early Holocene. It is widely accepted that the East Asian summer monsoon usually follows the variation of low-latitude summer solar radiation (Yuan et al., 2004; Chen et al., 2006; An et al., 2015). Charney (1969) and Wang (2009) also proposed that the seasonal migration of the intertropical convergence zone (ITCZ) profoundly influences the seasonality of the global monsoons. However, the global westerly winds and their associated storm tracks dominate the mid-latitude dynamics of the global atmosphere and affect the extratropical large-scale temperature and precipitation patterns (Oster et al., 2015; Voigt et al., 2015). Since the Last Glacial Maximum (LGM), climate in central and southern regions of the North American continent gradually dries out as the ice sheet melt and the westerlies move to north (Qin et al., 1997). As mentioned in the foregoing studies, millennial scale evolution in global monsoons and westerly winds probably shows different patterns as a result of complex driving mechanisms. Arguments about an asynchronous pattern of moisture variations between monsoon and westerly winds evolution underscore the importance of studying their millennial scale differentiation (Chen et al., 2006, 2008, 2019; An and Chen, 2009; Li et al., 2011; An et al., 2012).

A way to examine past climate variability is traditional methods of studying various archives which truly document the evolution of regional climate, including lake sediments (Madsen et al., 2008), stalagmites (Dykoski et al., 2005; Wang et al., 2008) and tree rings (Linderholm and Braeuning, 2006). However, due to the limited time scale of paleoclimate records, most researches on the evolution of monsoons and westerly winds are concentrated in the Holocene and lack an exploration during the LGM. With the development of paleoclimatology in recent decades, numerical simulations of paleoclimate continue to emerge and develop to a relatively mature system, which provides a useful tool for reviewing paleoclimate change over long time scales. On account of water balance system constantly responding to climatic conditions changes, a combination of numerical simulations and lake water balance models can be used to effectively track past climate change, and make up the deficiency in qualitative method of multi-proxy analysis (Qin and Yu, 1998; Xue and Yu, 2000; Morrill et al., 2001, 2004; Li and Morrill, 2010, 2013; Lowry and Morrill, 2019; Li et al., 2020). Covering one-fifth of the terrestrial surface, global closed basins distribute in both low-latitude monsoon regions and mid-latitude westerlies. Furthermore, closed basins with relative independent hydrological cycle system have a plenty of terminal lakes records that provide more evidence for retrospecting climate change (Li et al., 2017), and can be regarded as ideal regions for studying spatiotemporal difference between monsoons and westerly winds.

By constructing virtual lakes systems, here we applied lake models and a transient climate evolution model to continuously simulate water balance change since the LGM in global closed basins. Meanwhile, precipitation minus evaporation (P-E) simulations and 37 lake status records in the LGM, mid-Holocene (MH) and Pre-Industrial (PI) were supplemented for validating results of the continuous simulations. The prominent spatial differentiation of monsoons and westerly winds revealed by simulations leads us to focus on the Northern Hemisphere mid-latitude closed basins which are simultaneously influenced by mid-latitude westerly winds and low-latitude monsoons. In the mid-latitude closed basins of the Northern Hemisphere, the good match between water balance simulation and reconstructed moisture index from 27 paleoclimate records verifies the reliability of the simulation results. Further, we disaggregated the Northern Hemisphere

mid-latitude closed basins into the areas dominated by monsoons and westerly winds respectively, and emphatically explored the temporal evolution of the East Asian summer monsoon and westerly winds since the LGM. According to the climate records, we comprehensively considered the determinants that control the trend of climate change in the Northern Hemisphere westerlies and East Asian summer monsoon regions since the LGM. This study not only reveals millennial scale climate change from the perspective of water balance, but also provides a new method for studying the synergy of the westerly winds and monsoons.

## 2 Material and Methods

### 2.1 Experimental design

#### 2.1.1 Transient climate evolution experiment and CMIP5/PMIP3 multi-model ensemble

Transient climate evolution experiment (TraCE-21 kyr) as a synchronously coupled atmosphere-ocean circulation model simulation, is completed by the Community Climate System Model version 3 (CCSM3) (He, 2011). We applied this model to continuously simulating effective moisture change represented by virtual water balance variation since the LGM. Likewise, CCSM4, CNRM-CM5, FGOALS-g2, GISS-E2-R, MIROC-ESM, MPI-ESM-P and MRI-CGCM3 models participating in CMIP5/PMIP3 were also used to simulate the relative change of P-E during three particular periods (LGM, MH, PI). Here the PI period which is considered as a typical period of the late Holocene, is mainly used to measure the changes of hydroclimate conditions during the LGM and MH periods relative to the late Holocene, and verify the feasibility of the lake models by comparing the lake level simulations with the lake status records among three periods. PMIP3 protocols define the boundary conditions of these models, with a few exceptions (Table 1). Precession, obliquity and eccentricity values are specified according to Berger (1978). $CO_2$, $CH_4$, and $N_2O$ values are set on the basis of reconstructions from ice cores (Monnin et al., 2004; Flückiger et al., 1999, 2002). A remnant Laurentide ice sheet in the LGM and a modern-day ice sheet configuration in the MH and PI simulations are specified by the ICE-5G reconstruction (Peltier, 2004), while the vegetation is prescribed to modern values. Ice sheet configuration and vegetation distribution are used by GISS model. LGM radiative forcing changes in MIROC model and MRI model are the exceptions of the PMIP3 boundary conditions, details are shown in Licciardi et al. (1998) and Lowry and Morrill (2019).

**Table 1.** Boundary conditions in CMIP5/PMIP3 simulations at PI, MH and LGM.

|  | Pre-industrial | Mid-Holocene | Last Glacial Maximum |
|---|---|---|---|
| Eccentricity | 0.016724 | 0.018682 | 0.018994 |
| Obliquity (°) | 23.446° | 24.105° | 22.949° |
| Longitude of perihelion (°) | 102.04° | 0.87° | 114.42° |
| $CO_2$ (ppm) | 280 | 280 | 185 |
| $CH_4$ (ppb) | 760 | 650 | 350 |
| $N_2O$ (ppb) | 270 | 270 | 200 |

| | | | |
|---|---|---|---|
| Ice sheet | Peltier (2004) 0 ka | Peltier (2004) 0 ka | Peltier (2004) 21 ka |
| Vegetation | Present-day | Present-day | Present-day |

### 2.1.2 Lake energy balance model and lake water balance model

Before calculating, we linearly interpolated different resolutions grid cells of TraCE model and multi-model ensemble into a uniform resolution of 0.5°×0.5°. For all grid cells in closed basins, we assumed that the virtual lake in each grid cell is a 1 meter deep lake with freshwater, and then the virtual lake evaporation is calculated by a lake energy balance model that is modified according to Hostetler and Bartlein (1990)'s model. The evaporation of lake surface depends on the heat capacity of water, water density, lake depth, lake surface temperature, shortwave radiation, longwave radiation absorbed by the water surface, longwave radiation emitted by the water surface, latent heat flux, and sensible heat flux, etc. If the surface energy balance is negative (positive), the ice forms (melts). Besides, lake depth and lake salinity are important input parameters influencing lake surface evaporation (Dickinson et al., 1965), however, only small changes appear in lake evaporation when adding lake depth to 5 and 10 m and increasing lake salinity to 10 ppt. More details of lake energy balance model are described in Morrill (2004) and Li and Morrill (2010).

For better assessing the relative change of water balance since the LGM, the virtual lakes are assumed in hydrological equilibrium with steady state. The lake water balance equation is shown as follows:

$$D = A_B R + A_L(P_L - E_L) , \quad (1)$$

where $D$ is discharge from the lake (m$^3$ year$^{-1}$), $A_B$ is area of the drainage basin (m$^2$), $R$ is runoff from the drainage basin (m year$^{-1}$), $A_L$ is area of the lake (m$^2$), $P_L$ is precipitation over the lake (m year$^{-1}$) and $E_L$ is lake evaporation (m year$^{-1}$). Given the application of Equation (1) requiring specific values of the $A_B$ and $A_L$, this equation is simplified for grid cells where $P_L - E_L \geq 0$ and grid cells where $P_L - E_L < 0$. Grid cells where $P_L - E_L \geq 0$ represent open lakes, and maintain water balance by discharging more or less water. While the runoff into the lake compensates the net water loss in grid cells where $P_L - E_L < 0$, and these regions maintain water balance by changes in the ratio of $A_L$ to $A_B$, as described by setting $D = 0$ in Equation (2):

$$\frac{A_L}{A_B} = \frac{R}{(E_L - P_L)} , \quad (2)$$

where $A_L/A_B$ represents virtual lake level. Accordingly, for grid cells with $P_L - E_L < 0$, the $A_L/A_B$ values are calculated and compared to represent relative water balance change, and more details about lake water balance model are described in Li and Morrill (2010). We combined the values of $P_L$, $E_L$ and $R$ with Equations (1) and (2) and simulated the continuous water balance change since the LGM using TraCE 21 kyr model.

### 2.2 Records selection and moisture index inference

lake status information in or near global closed basins were collected to compare relative changes among three characteristic periods. Lake status information sorted by latitudes is shown in Table 2. Then, 27 climate records were compiled in or near the mid-latitude closed basins of the Northern Hemisphere with reliable chronologies and successive

sedimentary sequences from published literatures, which can reflect the continuous dry and wet change (Table 3). We
interpolated climate data at intervals of 10 years and unified the time scale according to the chronology accuracy of the
extracted data. Finally, the data were standardized to indicate a humid climate with a relative high value and a dry climate
with a relative low value, and the signals of moisture change were transformed into a range of 0 to 1 index. Due to the
different time scales of the collected continuous paleoclimate records, we can only reconstruct the regional moisture change

125    from the early to late Holocene after unifying the time scales, but the purpose of this part is only to check the simulation
results.

**Table 2.** Summary of lake level change in or near global closed basins.

| Lake | Location | Lat(°) | Lon(°) | Materials and dating methods | LGM relative to MH | LGM relative to PI | MH relative to PI | References |
|---|---|---|---|---|---|---|---|---|
| Achit Nuur | Mongolia | 49.42 | 90.52 | Sediments and AMS [14]C | High | High | High | Sun et al., 2013 |
| Ulungur Lake | China | 46.98 | 87 | Sediments and AMS [14]C | Low | Low | High | Mischke et al., 2011 |
| Manas Lake | China | 45.75 | 86 | Sediments and AMS [14]C | Low | Low | Low | Rhodes et al., 1996 |
| Ebinur Lake | China | 44.9 | 82.7 | Sediments and OSL dating | High | High | High | Wu et al., 1995; Jin et al., 2013 |
| Lower Red Rock Lake | America | 44.63 | -111.84 | Sediments and AMS [14]C | High | High | High | Mumma et al., 2012 |
| Balikun Lake | China | 43.67 | 92.8 | Sediments and U–Th dating | High | High | High | Ma et al., 2004; Lu et al., 2015 |
| Bosten Lake | China | 42 | 87 | Sediments and AMS [14]C | Low | Low | High | Wünnemann et al., 2006; Huang et al., 2009 |
| Surprise Lake | America | 41.5 | -120.1 | Sediments and U–Th dating | High | High | Similar | Ibarra et al., 2014 |
| Bonneville Lake | America | 40.5 | -113 | Terraces and [14]C | High | High | Low | Oviatt, 2015; Hart et al.,2004 |
| Yitang Lake | China | 40.3 | 94.97 | Sediments and OSL dating | Low | Low | High | Zhao et al., 2015 |
| Lop Nur Lake | China | 40.29 | 90.8 | Sediments and U–Th dating | High | High | High | Yan et al., 2000 |
| Yanhaizi Lake | China | 40.1 | 108.42 | Sediments and AMS [14]C | High | High | Similar | Chen et al., 2003 |
| Lahontan Lake | America | 40 | -119.5 | Terraces and [14]C | High | High | High | Lyle et al., 2012 |
| Qingtu Lake | China | 39.05 | 103.67 | Terraces and AMS [14]C | High | High | Similar | Zhang et al., 2004 |
| Karakul Lake | Tajikistan | 39.02 | 73.53 | Sediments and AMS [14]C | Low | High | High | Heinecke et al., 2017 |
| Van Lake | Turkey | 38.5 | 43 | Sediments and AMS [14]C | High | High | High | Çağatay et al., 2014 |
| Hala Lake | China | 38.2 | 97.4 | Sediments and AMS [14]C | Low | Low | Low | Yan and Wünnemann, 2014 |
| Owens Lake | America | 38 | -119 | Terraces and [14]C | High | High | / | Bacon et al., 2006 |
| Qinghai Lake | China | 36.53 | 99.6 | Terraces and AMS [14]C | Low | Low | Similar | Madsen et al., 2008 |
| Bangong Co | China | 33.7 | 79 | Sediments and AMS [14]C | Similar | High | High | Rossit et al., 1996; Li et al., 1991 |
| Cochise Lake | America | 32.1 | -109.8 | Sediments and [14]C | High | High | High | Waters, 1989 |
| Cloverdale Lake | America | 31.5 | -109 | Terraces and [14]C | High | High | / | Krider, 1998 |
| Zabuye Lake | China | 31.35 | 84.07 | Sediments and AMS [14]C | High | High | High | Wang et al.,2002 |
| Nam Co | China | 30.65 | 90.5 | Sediments and AMS [14]C | Low | Low | High | Witt et al., 2016 |
| Babicora Lake | Mexico | 29 | -108 | Sediments and U–Th dating | High | High | / | Metcalfe et al., 2002 |
| Chen Co | China | 28.93 | 90.6 | Sediments and AMS [14]C | High | Similar | High | Zhu et al., 2009 |
| La Piscina de Yuriria Lake | Mexico | 20.22 | -100.13 | Sediments and [14]C | Low | Low | High | Davies, 1995 |

| Chignahuapan Lake | Mexico | 19.16 | -99.53 | Sediments and $^{14}$C | High | High | / | Caballero et al., 2002 |
|---|---|---|---|---|---|---|---|---|
| Pátzcuaro Lake | Mexico | 19.5 | -101.5 | Sediments and AMS $^{14}$C | High | High | Low | Bradbury, 2000 |
| Malawi Lake | Malawi | -10.02 | 34.19 | Sediments and OSL dating | Low | Low | High | Konecky et al., 2011 |
| Titicaca Lake | Peru/Bolivia | -16 | -69.4 | Sediments and AMS $^{14}$C | High | High | Low | Rowe et al., 2002 |
| Makgadikgadi Lake | Botswana | -20 | 24.76 | Terraces and $^{14}$C | High | High | High | Riedel et al., 2014 |
| Uyuni Lake | Bolivia | -20.2 | -67.5 | Sediments and U–Th dating | High | High | High | Baker et al., 2001 |
| Mega-Frome Lake | Australia | -31 | 140 | Terraces and AMS $^{14}$C | High | High | High | Cohen et al., 2011 |
| Cari Laufquen Lake | Argentina | -41.4 | -69.6 | Sediments and $^{14}$C | / | High | / | Cartwright et al., 2011 |
| Huelmo Lake | Chile | -41.5 | -73 | Sediments and AMS $^{14}$C | High | High | / | Moreno and León, 2003 |
| Potrok Aike Lake | Argentina | -52 | -70.4 | Sediments and OSL dating | High | High | / | Kliem et al., 2013 |

**Table 3.** Paleoclimatic records indicating dry or wet status.

| Lake | Location | Lat (°) | Lon (°) | Elevations (m) | Dating method | Resolution (yr) | Dates number | Time period (cal yr BP) | Proxies used | References |
|---|---|---|---|---|---|---|---|---|---|---|
| Karakul Lake | Tajikistan | 39.02 | 73.53 | 3915 | $^{14}$C | ~200 | 5 | 10000-0 | TOC, TOC/TN, δ18Ocarb, TIC | Heinecke et al., 2017 |
| Achit Nuur | Mongolia | 49.42 | 90.52 | 1444 | AMS $^{14}$C | ~220-440 | 10 | 22000-0 | Pollen | Sun et al., 2013 |
| Ulungur Lake | China | 46.98 | 87 | 478.6 | AMS $^{14}$C | ~40 | 6 | 10000-0 | grain-size, pollen data | Liu et al., 2008 |
| Lower Red Rock Lake | America | 44.63 | -111.84 | 2015 | AMS $^{14}$C | ~410 | 5 | 22000-0 | Magnetic susceptibility, Carbonate, Organic | Mumma et al., 2012 |
| Ulaan Nuur | Mongolia | 44.51 | 103.65 | 1110 | OSL | ~60 | 12 | 16000-0 | TOC, TN, C/N, CaCO3, CIA | Lee et al., 2013 |
| Jenny Lake | America | 43.76 | -110.73 | 2070 | AMS $^{14}$C | ~200 | 11 | 14000-0 | TOC, C/N | Larsen et al., 2016 |
| Balikun Lake | China | 43.67 | 92.8 | 1580 | $^{14}$C | ~30 | 7 | 10000-0 | TOC, δ18Ocar | Xue et al., 2011 |
| Lake Woods | America | 43.48 | -109.89 | 2816 | $^{14}$C | ~120 | 17 | 12000-0 | Sand content | Pribyl and Shuman, 2014 |
| Blue Lake | America | 40.5 | -114.04 | 1297 | AMS $^{14}$C | ~280 | 12 | 14000-1000 | Pollen | Louderback and Rhode, 2009 |
| Yitang Lake | China | 40.3 | 94.97 | / | OSL | ~110 | 4 | 23000-0 | TOC, C/N, δ13Corg | Zhao et al., 2015 |
| Tiao Lake | China | 40.26 | 99.31 | 1188 | AMS $^{14}$C | ~195 | 4 | 11000-1000 | Rb/Sr, Fe/Mn | Li et al., 2013 |
| Yanhaizi Lake | China | 40.1 | 108.42 | 1180 | $^{14}$C | ~80 | 17 | 14000-0 | TOC, magnetic susceptibility, maturity index | Chen et al., 2003 |
| Yanchi Lake | China | 39.72 | 99.17 | 1200 | AMS $^{14}$C | ~250 | 14 | 18000-0 | TOC, C/N, Carbonate | Li et al., 2013 |
| Qingtu Lake | China | 39.05 | 103.67 | 1309 | AMS $^{14}$C | ~40 | 11 | 11000-0 | C/N, grain size | Li et al., 2012 |
| Van Lake | Turkey | 38.5 | 43 | 1649 | AMS $^{14}$C | ~200 | 3 | 25000-0 | TOC, TIC, δ13C, δ18O | Öğretmen.,2012 |
| Hala Lake | China | 38.2 | 97.4 | 4078 | $^{14}$C | ~150 | 18 | 24000-0 | OM, Carbonate | Yan et al., 2014 |
| Sanjiaocheng | China | 39.01 | 103.34 | 1325 | AMS $^{14}$C | ~50 | 11 | 15000-0 | TOC, δ13Corg | Zhang et al., 2004 |
| Hurleg Lake | China | 37.28 | 96.9 | 2817 | AMS $^{14}$C | / | 8 | 10000-0 | Carbonate | Zhao et al., 2010 |
| Gahai Lake | China | 37.13 | 97.55 | 2850 | AMS $^{14}$C | ~90 | 27 | 12000-0 | δ13Cc, δ18Oc, CaCO3 | Guo et al., 2012 |
| Chaka Lake | China | 36.63 | 99.03 | 3200 | AMS $^{14}$C | / | 10 | 10000-0 | TOC, TN | Liu et al., 2008 |
| Qinghai Lake | China | 36.53 | 99.6 | 3200 | AMS $^{14}$C | ~30 | 10 | 18000-0 | TOC, TN, C/N, Carbonate | Shen et al., 2005 |
| Dalianhai Lake | China | 36.24 | 100.39 | 2852 | $^{14}$C | ~10 | 28 | 24000-0 | Rb/Sr | Wu, 2017 |
| Zigetang Co | China | 32 | 90.73 | 4560 | $^{14}$C | / | 5 | 10500-0 | TOC, TOC/TS, HI, δ13Corg, TC, TIC | Wu et al., 2007 |
| Bangong Co | China | 33.7 | 79 | 4241 | AMS $^{14}$C | ~80 | 11 | 10000-0 | δ18O | Fontes et al., 1996 |
| Zabuye Lake | China | 31.35 | 84.07 | 4421 | AMS $^{14}$C | ~620 | 17 | 30000-0 | TOC, TIC, δ18Ocarb, δ13Ccarb | Wang et al., 2002 |

| | | | | | | | | | | |
|---|---|---|---|---|---|---|---|---|---|---|
| Gonghai Lake | China | 38.9 | 112.23 | 1860 | AMS $^{14}$C | ~15 | 25 | 14700-0 | Pollen | Chen et al., 2015 |
| Dali lake | China | 43.15 | 116.29 | 1220 | AMS $^{14}$C | ~350 | 27 | 16000-0 | Lake elevation | Goldsmith et al., 2016 |

## 2.3 Mathematical methods

Linear tendency estimation is a common trend analysis method, which was chosen to measure the variation degree of simulated water balance in this paper. Besides, we also used the Empirical orthogonal function (EOF), a method of analyzing the structural features in matrix data and extracting the feature vector of main data, to examine spatially and temporally variability of simulated water balance. The spatial distribution of EOF first (second) mode is denoted by EOF1 (EOF2), and the time series of first (second) mode is denoted by PCA1 (PCA2).

## 3 Results and discussion

### 3.1 Observed and simulated water balance change in global closed basins

As Fig. 1 shown, we intercepted LGM (18000-22000 yr), MH (5000-7000 yr) and PI (1800-1900 AD) periods from the TraCE 21 kyr dataset for better matching the multi-model ensemble. Because runoff anomalies are highly correlated to precipitation anomalies, it is therefore feasible to consider that the contribution of runoff on water balance is considered as the contribution of precipitation on water balance. Difference between the time period we chosen subjectively and the time periods defined by the multi-model ensemble may affect the comparison results. However, precipitation and evaporation difference of TraCE 21 kyr among three periods exhibits similar spatial pattern with P-E difference of multi-model ensemble. The simulations and lake status records of the mid-latitude westerlies (low-latitude monsoon regions) show that LGM is humid (dry) relative to MH and PI, which generally corresponds to the hydroclimate patterns of previous researches (Street and Grove, 1979; Qin et al., 1997; Quade and Broecker, 2009; Lowry and Morrill, 2019). It's not our intent to simulate relative lake status change among three periods, but to validate continuous water balance simulations and to track continuous water balance fluctuations on the millennial scale using TraCE 21 kyr simulations.

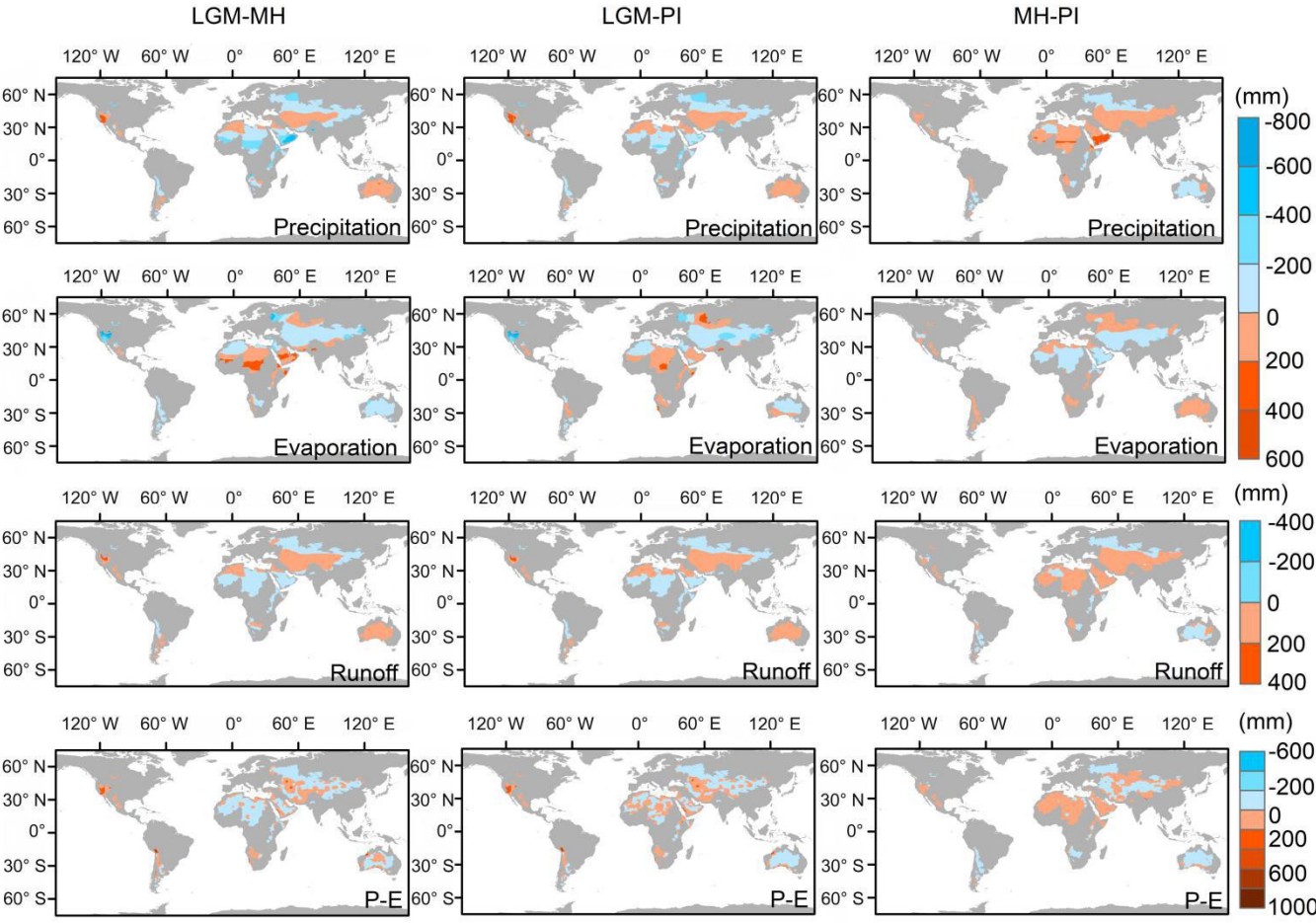

**Figure 1.** Annual mean precipitation, evaporation and runoff from TraCE 21 kyr simulations, and precipitation minus evaporation (P-E) from multi-model ensemble, all units mm year$^{-1}$; (first column) difference between LGM and MH simulations; (second column) difference between LGM and PI simulations; (third row) difference between MH and PI simulations.

In continuous simulations, we partitioned the trend map of water balance into positive and negative components to highlight the spatial patterns of water balance change (Fig. 2). In the global mid-latitude westerlies, simulations indicate widespread effective moisture declines since the LGM except the northern Caspian Sea, whereas, effective moisture increases since the LGM over the global Tropics. Meanwhile, the trend map exactly exhibits the spatial differentiation of the millennial scale water balance change between the global low-latitude monsoon dominated regions and the mid-latitude westerly winds dominated regions. This differentiation provides the basis to explore the continuous evolution of monsoons and westerly winds in the closed basins since the LGM.

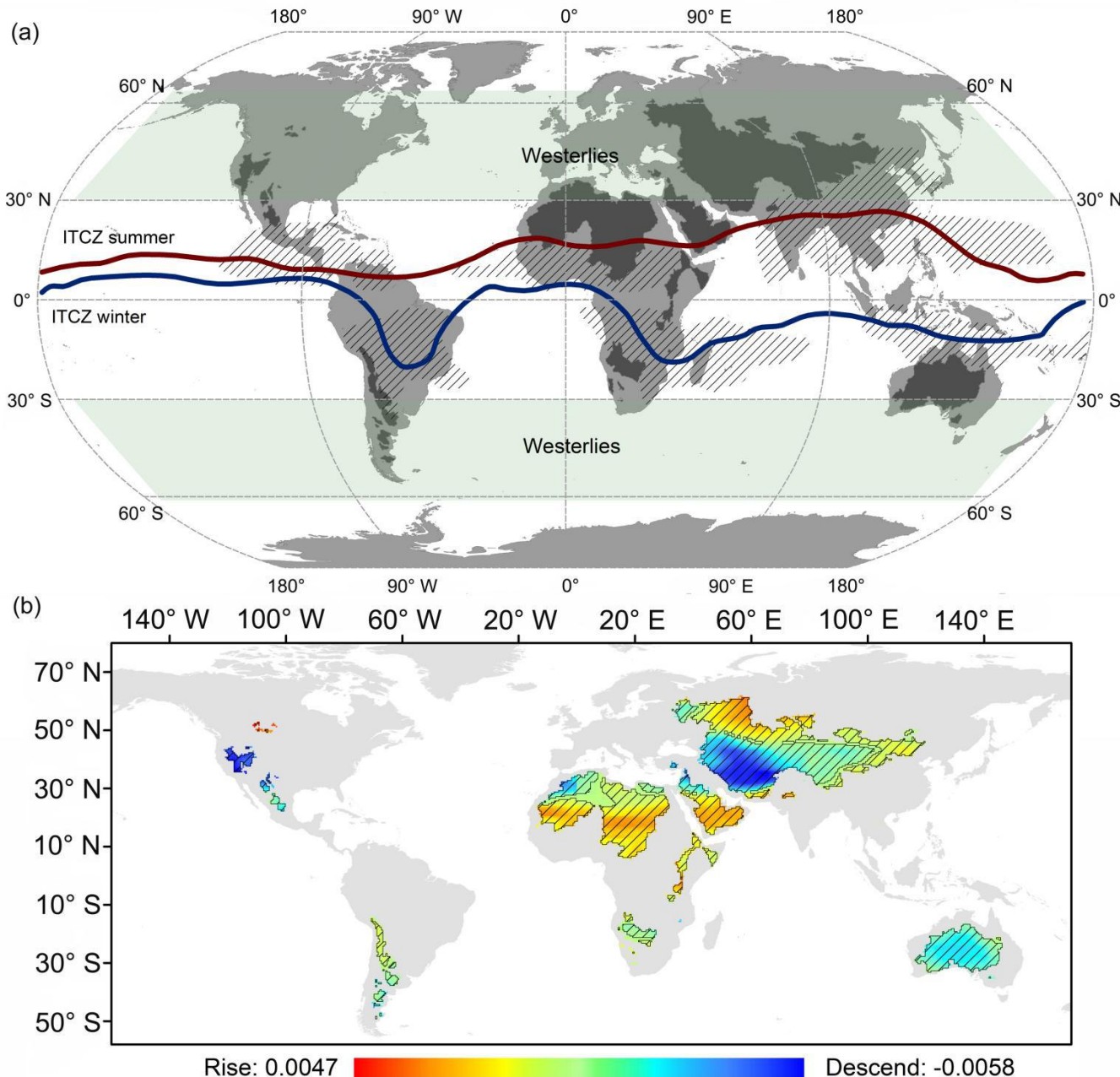

**Figure 2.** (a) Distribution of global closed basins and circulation system: The dark areas are global closed basins; summer and winter of the ITCZ are in accordance with the Northern Hemisphere; the shadows present the six monsoon areas according to Wang (2009), and (b) Trend analysis of continuous simulation in water balance change: The shadows indicate that the trends are statistically significant at 5% level.

## 3.2 Possible driving mechanisms of millennial scale water balance change

In this section, the possible driving mechanism that affects the millennial scale water balance change in the global closed basins is explored. Positive signs of the EOF1 represent most monsoon regions of mid-latitudes and low-latitudes, while negative signs of that are mainly located in the Northern and Southern Hemisphere westerlies. Spatial characteristics of the EOF2 have an opposite trend with the EOF1, except for the Caspian Sea. The contribution rate of PCA1 and PCA2 is 51% and 14% respectively, therefore the following discussion mainly focuses on PCA1 with the high contribution rate (Fig. 3). The PCA1 extracted from water balance simulation tends to represent the effective moisture fluctuation of closed basins in low-latitude monsoon regions, indicating a relative humid climate during the early-to-mid Holocene. By comprehensively analyzing a variety of paleoclimate proxies, Wang et al. (2017) suggested that moisture change revealed by the Australian monsoon, the East African monsoon and the Indian monsoon regions reaches the wettest status in the early Holocene, while the wettest condition in the East Asian summer monsoon regions occurs between 8 and 6 kyr. Likewise, Qin (1997) presented that the wettest period in the African and South Asian monsoon regions is the early-to-mid Holocene, coinciding well with our results.

The climatic significance of the $\delta^{18}O$ in the Asian speleothem records is always a long-standing debate, and some influential hypotheses regard $\delta^{18}O$ of the monsoon regions as a proxy for "Asian monsoon intensity", "Indian monsoon intensity", "summer monsoon rainfall amount" and "circulation conditions" (Cheng et al., 2012; Chen et al., 2016). Although the climatic significance is controversial, it is well accepted that $\delta^{18}O$ changes should bear the imprint of variations in the oxygen isotopic composition of precipitation (Cheng et al., 2012; Chen et al., 2016). According to the close similarity of the PCA1 with the speleothem records from Dongge and Hulu caves, our simulations are more inclined to suggest that the $\delta^{18}O$ stalagmite records indicate the change in water vapor brought by the monsoons. In addition, we not only compared the PCA1 with the stalagmite records of Dongge Cave with controversial climatic significance, but also with the summer solar radiation at low-latitudes in the Northern Hemisphere. This comparison provides evidence for the view that the evolution of low-latitude monsoons is generally controlled by summer insolation in the Northern Hemisphere (Yuan et al., 2004; Chen et al., 2006; An et al., 2015). Thus, we further speculated that the water balance change in monsoon regions of global closed basins is mainly driven by mid-latitude and low-latitude summer solar radiation.

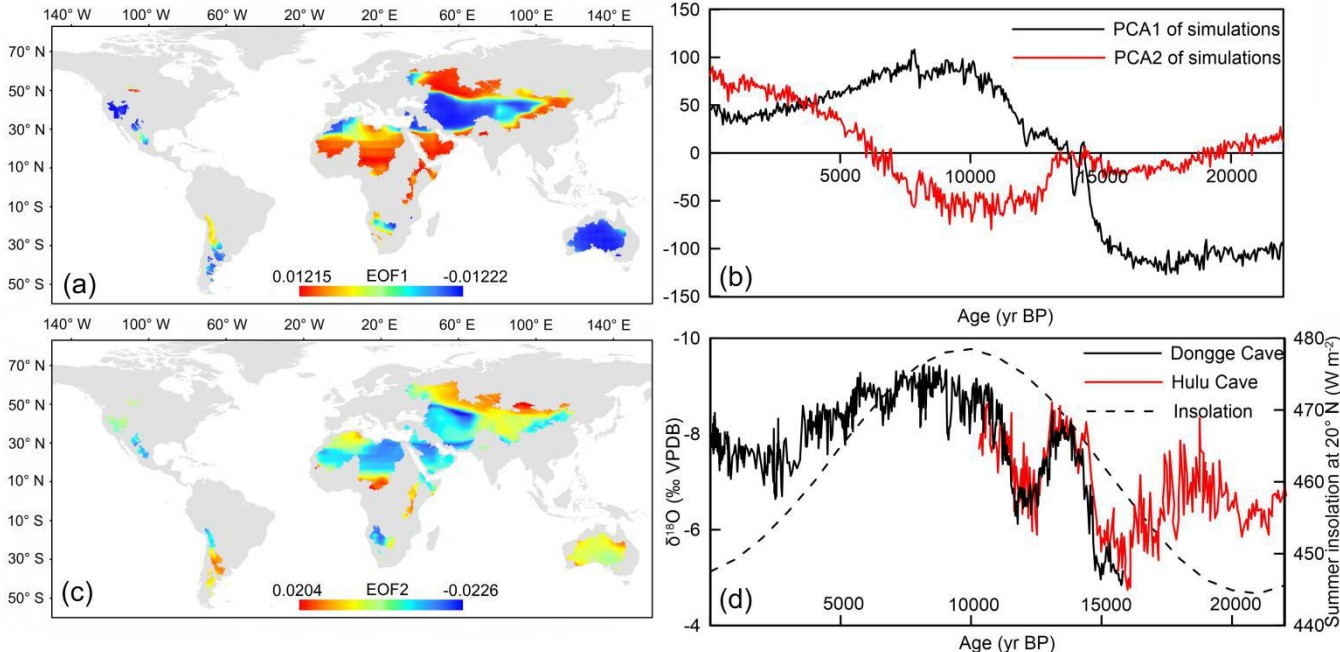

**Figure 3.** (a) Spatial distribution feature of EOF1, (b) PCA1 and PCA2 of simulated water balance change since the LGM, (c) Spatial distribution feature of EOF2, and (d) Comparison between stalagmite records and summer insolation: Stalagmite records come from Dykoski et al. (2005) and Wang et al. (2008), and summer insolation comes from Berger (1978).

## 3.3 Evolutionary characteristics and causing factors of millennial scale hydroclimate change in the Northern Hemisphere mid-latitude closed basins

On the basis of the spatial characteristics of the EOF analysis, closed basins in the Northern Hemisphere, affected both by low-latitude monsoons and mid-latitude westerly winds, are ideal regions for revealing synergy of the westerly winds and monsoons. Between 30°N and 60°N, 27 paleoclimate records indicating dry or wet climate were collected from the Northern Hemisphere mid-latitude closed basins. As described in Sect. 2.2, we reconstructed moisture index from the early to late Holocene around that regions (Fig. 4). Simulated mean water balance curve corresponds well with mean moisture index in the Northern Hemisphere mid-latitude closed basins, indicating a transition from a humid climate in the early-to-mid Holocene to an arid climate in the late Holocene. Therefore, continuous simulations, well validated by the paleoclimate indicators, could be better used to track climate change during the LGM.

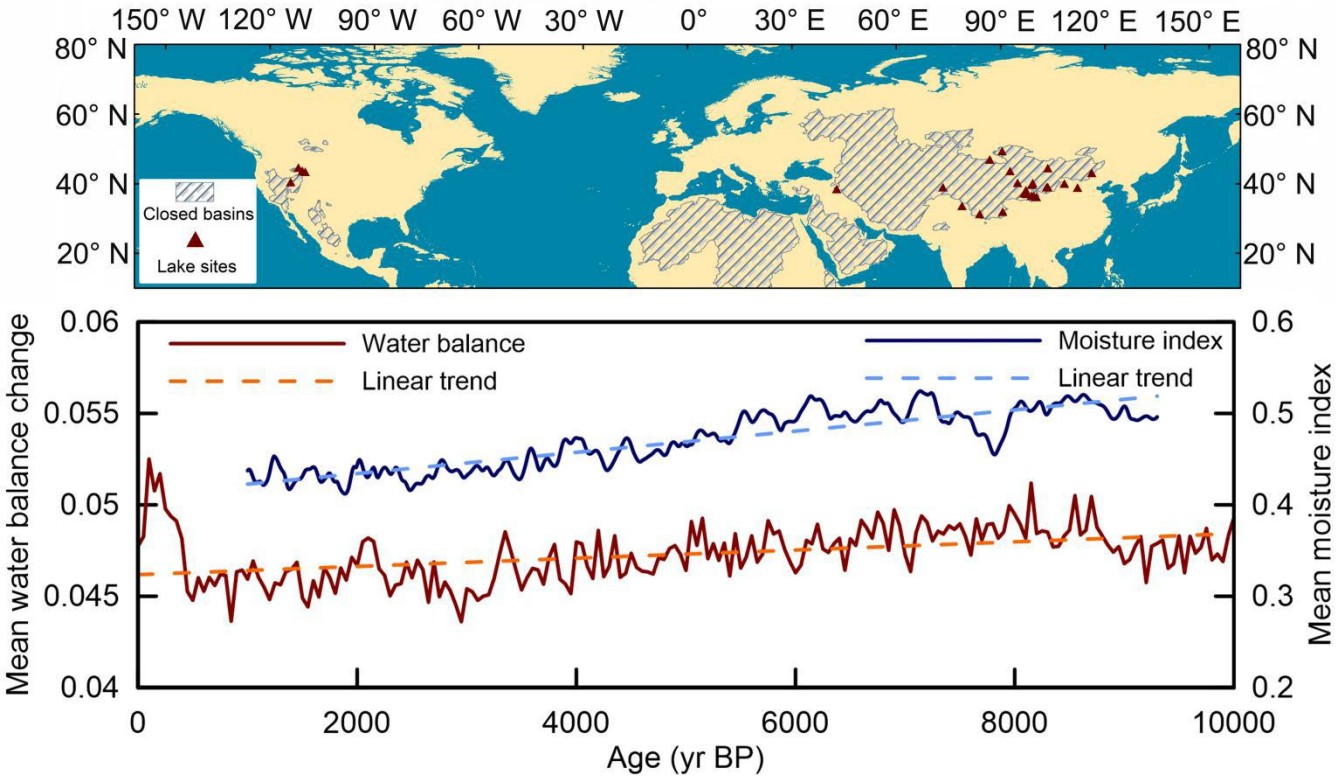

**Figure 4.** Comparison between simulated water balance change and reconstructed moisture index in the mid-latitude closed basins of the Northern Hemisphere during the Holocene. Triangles indicate locations of paleoclimate records (Table 3).

Water balance simulations since the LGM show that a humid climate not only appears in the early-to-mid Holocene but also occurs during the LGM, while the climate is relatively dry in the late Holocene. The maintained high moisture in the LGM is possibly influenced by low evaporation and high precipitation (Fig. 5). Using paleoclimate modelling, Yu et al. (2000) mentioned that the low temperature during the glacial period causes a decrease of evaporation and a reduction of lake water loss, resulting in the appearance of high lake level. Afterward, solar radiation, atmosphere radiation, temperature, evaporation and precipitation simulations gradually increase (Fig. 5). When entering the warm Holocene, precipitation continues increasing and reaches a maximum in the MH, while solar radiation, atmosphere radiation and evaporation decrease during the early-to-mid Holocene and then increase around the late Holocene. Low (high) evaporation and high (low) precipitation are responsible for the MH (late-Holocene) relative humid (dry) climate (Fig. 5).

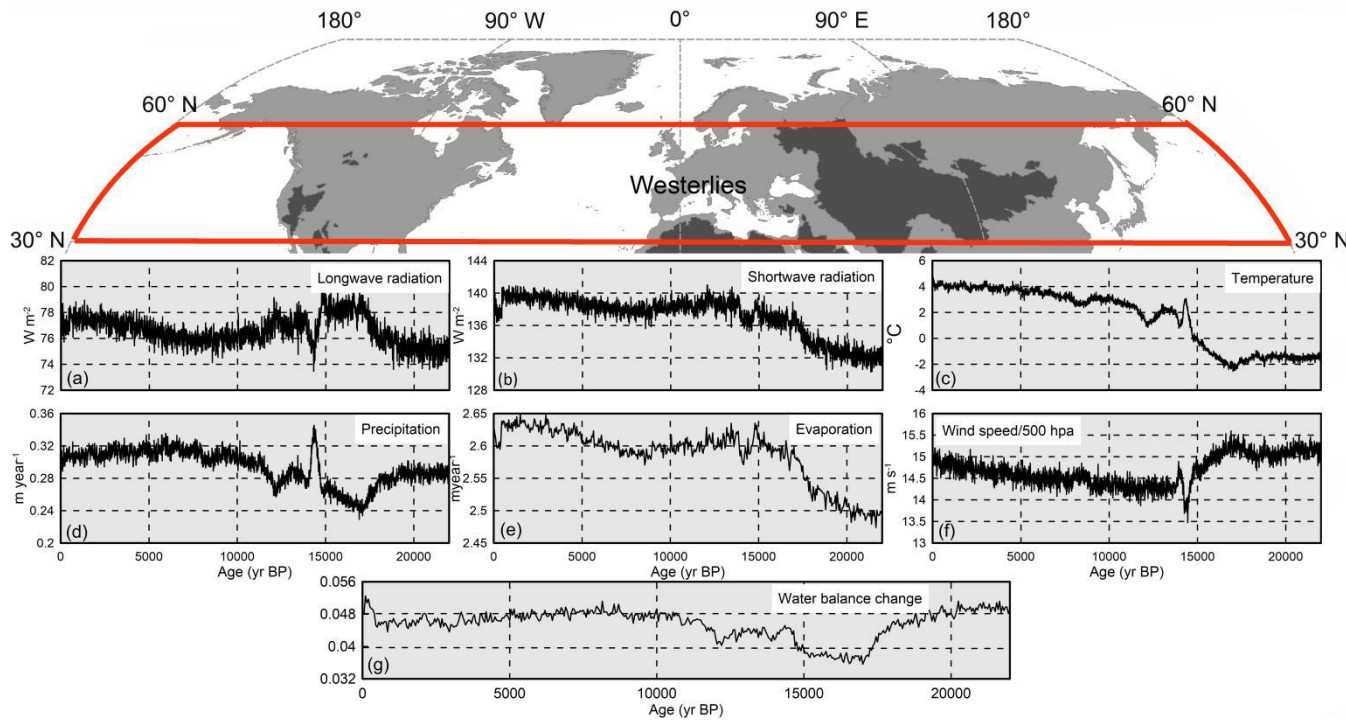

**Figure 5.** Time series of (a) longwave radiation, (b) shortwave radiation, (c) temperature, (d) precipitation, (e) evaporation, (f) 500 hpa wind speed and (g) water balance change between 30°N and 60°N closed basins since the LGM.

### 3.4 Evolutionary differentiation of millennial scale monsoons and westerly winds in Asian closed basins

Spatial distributions of the EOF1 and EOF2 clearly exhibit that a prominent boundary exists in the interactional zones
between East Asian summer monsoon and westerly winds in Asia. Since the boundary of the monsoon will be adjusted accordingly with the change of East Asian summer monsoon strength, evolution of Asian lakes on the millennial scale probably not follows a single climate changing pattern (Wu et al., 2000; Editorial Committee of China's Physical Geography, 1984; An et al., 2012). The regions dominated by East Asian summer monsoon and westerly winds were then selected respectively based on the spatial characteristics of two modes extracted from the EOF, to explore millennial scale evolution
features of two climate systems (Fig. 6). In the westerly winds dominated regions, the LGM and MH are characterized by humid climate, and relative dry climate prevails in the early and late Holocene. Whereas, the water balance in the monsoon dominated regions is generally affected by East Asian summer monsoon which brings much water vapor over the early-to-mid Holocene, and leads to relative dry climate in the LGM and late Holocene. Li (1990) first proposed the "monsoon" and "westerly" modes on the millennial scale since the late Pleistocene in northwest China, then different climate
changing patterns between arid central Asia and monsoonal Asia were demonstrated by numerous paleoclimate records (Chen et al., 2006, 2008; An and Chen, 2009; Li et al., 2011; Chen et al., 2019). Thereinto, a viewpoint that millennial scale

East Asian summer monsoon change is possibly driven by summer insolation change in low-latitudes is the most widely accepted (Yuan et al., 2004; Dykoski et al., 2005; Hu et al., 2008; Fleitmann et al., 2003). And the sea-surface temperatures (SSTs) of North Atlantic and air temperatures of high-latitudes are responsible for the Holocene effective moisture evolution of arid Central Asia which is dominated by the westerly winds (Chen et al., 2008). The moisture transport in the arid Central Asia mainly comes from the Northern Hemisphere westerlies of which the moisture source derives from the Black Sea, the Mediterranean Sea, the Arctic Ocean and the Atlantic Ocean. Winter precipitation accounts for a large proportion of annual precipitations in these regions (Li et al., 2008).

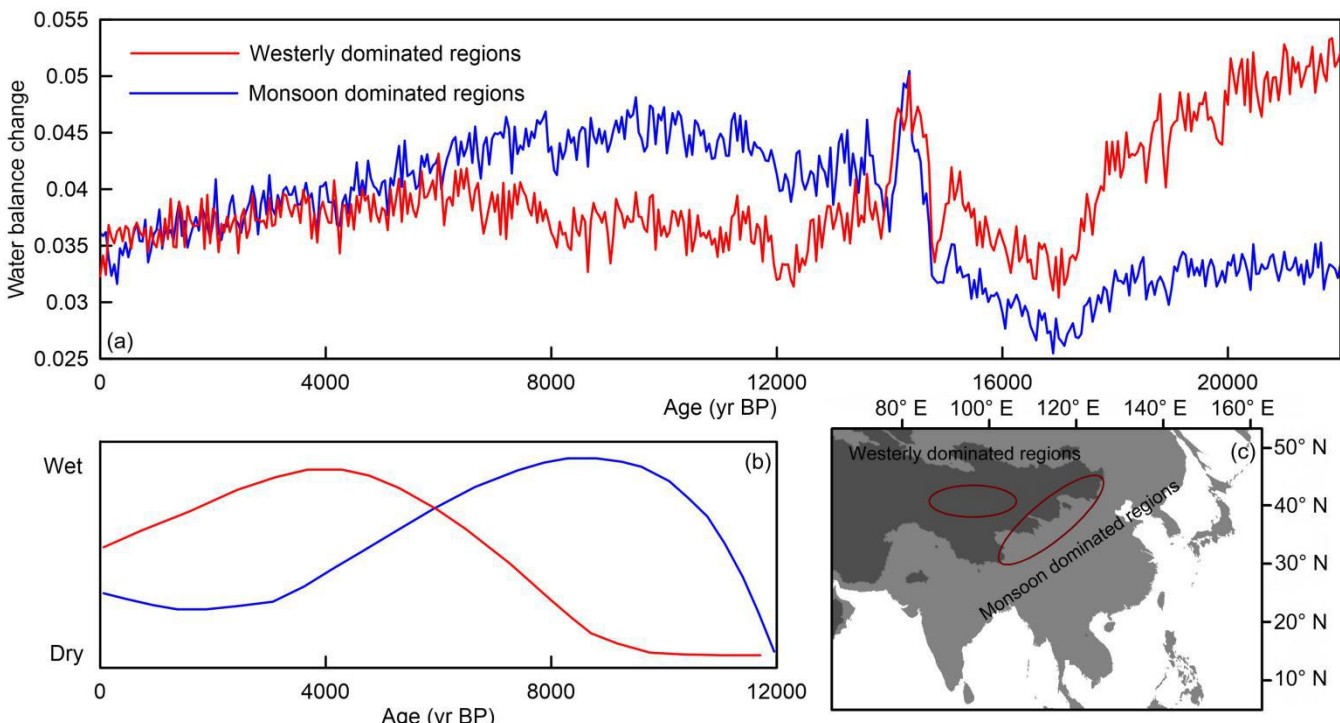

**Figure 6.** (a) Simulated water balance change between westerly dominated regions and monsoon dominated regions in the Asian closed basins since the LGM, (b) General climate changing patterns during the Holocene in monsoon Asia and westerly Central Asia come from Chen et al. (2006), and (c) Extracted westerly dominated regions and monsoon dominated regions in the Asian closed basins.

The water balance change in the Asian monsoon regions we extracted largely represents the hydroclimate variation in East Asian summer monsoon dominated regions since the LGM, while the water balance change in the westerly regions in Central Asia can represent the hydroclimate variation in the entire Northern Hemisphere westerlies. Qin (1997) made a large-scale spatial analysis and presented that lake levels in south-central North America change from high to low since the LGM and reach the lowest in early-to-mid Holocene. The LGM proxies indicate the southwestern America experienced a

climate that was wetter than present, and the Pacific Northwest through the Rockies experienced a climate that was drier than present, as well as a transition from wetter to drier conditions happened along a northwest-southeast trending band across the northern Great Basin (Oster et al., 2015). Our results generally reflect that the climate of westerlies is relatively wet at the LGM and relatively dry at the MH. For the Asian tropics in the Northern Hemisphere, the increased summer solar radiation from 12000 to 6000 yr induces the enhancement of thermal contrast between land and sea, and further causes the strengthening of summer monsoons, so that more water vapor is brought (COHMAP Members, 1988). Collected records in the Northern Hemisphere indicate evolution of westerly winds and monsoon systems (Fig. 7). Speleothem records from central and southern China confirm that the periods of weak East Asian summer monsoons are coincided with the cold periods of the North Atlantic (Yuan et al., 2004, Dykoski et al., 2005; Wang et al., 2008). The longest and highest-resolution drill core from Lake Qinghai (An et al., 2012) indicates that the summer monsoon record generally resembles the changing trends of Asian summer monsoon records derived from Dongge and Hulu speleothems over the last 20 kyr, and the mid-latitude westerlies climate dominates the Lake Qinghai area in glacial times. Low-latitude summer insolation is broadly recognized as a major control on low-latitudes monsoon systems, as a result, the tropical monsoons are weak during the LGM and late Holocene, and strong monsoons prevail in the early-to-mid Holocene (Fig. 7). Accordingly, the intensity of monsoon systems and westerly winds varies in different periods so that the main control system in the interactional regions depends largely on which system is much stronger during that period.

The Northern Hemisphere westerlies shifting northward or southward has a significant impact on global atmosphere circulation and inevitably affects the monsoon systems. Quaternary ice sheets of the Northern Hemisphere in the LGM develop to its maximum extension, and consequent existence of persisting strong glacial anticyclone leads to the southward displacement of the westerly winds (Yu et al., 2000). Many researches suggested the Northern Hemisphere westerlies in the LGM move to the southwest of the United States and the eastern Mediterranean region (Lachniet et al., 2014; Rambeau, 2010). Therefore, the narrowed temperature difference between sea and land causes the East Asian summer monsoon weaken, and may further induces the strong westerly winds throughout the year and then the precipitation increases (Yu et al., 2000). Furthermore, a growing body of evidence shows that the position and orientation of the westerly jet (WJ) probably control the Holocene East Asian summer rainfall patterns. A link between the northward seasonal progression of the WJ and the spatial pattern of East Asian summer monsoon precipitation shows that earlier northward progression of the WJ causes abundant precipitation at high-latitudes and less precipitation at low-latitudes (Nagashima et al., 2013). Especially the northward evolution of the WJ from south of the Tibetan Plateau and seasonal transition exert great influences on East Asian paleoclimate change (Chiang et al., 2015). Herzschuh et al. (2009) proposed that the position of summer monsoon rain band changes as the WJ axis shifts gradually southward, leading to the occurrence of spatiotemporal difference in Holocene China's maximum precipitation. In summary, the above views emphasize that the complex interaction between the monsoon and the westerly systems on the millennial scale should receive more attention.

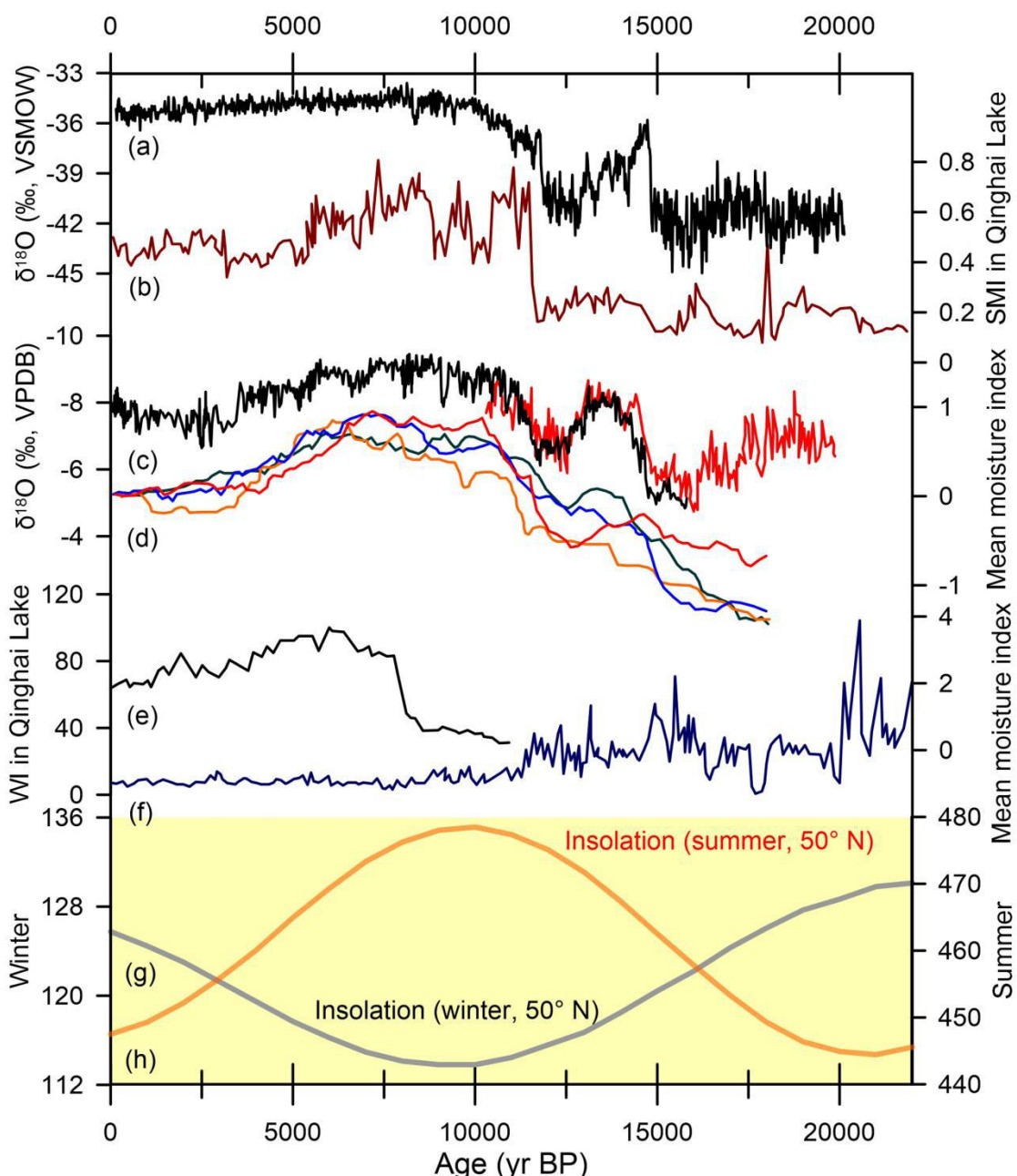

**Figure 7.** Comparison of records between the westerly and monsoon regions of the Northern Hemisphere. (a) NGRIP $\delta^{18}O$ (Rasmussen et al., 2006); (b) Lake Qinghai Westerlies climate index (An et al., 2012); (c) Dongge and Hulu cave speleothem $\delta^{18}O$ records (Dykoski et al., 2005; Wang et al., 2008); (d) moisture indexes in East Asian Monsoon (red line), East African Monsoon (green line), Indian Monsoon (blue line) and Australian Monsoon (orange line) regions (Wang et al., 2017); (e) The average moisture index for arid central Asian region as a whole during the Holocene (An and Chen, 2009); (f) Lake Qinghai

Asian summer monsoon index (An et al., 2012); (g) and (h) are winter 50°N insolation and summer 50°N insolation, respectively (Berger, 1978).

**4 Conclusion**

On the basis of 37 lake status records near global closed basins and 27 paleoclimatic records near mid-latitude closed basins of the Northern Hemisphere, we applied a lake energy balance model, a lake water balance model and paleoclimate simulations to exploring the millennial scale differentiation between global monsoons and westerly winds. Water balance simulation shows that the effective moisture in most closed basins of the Northern Hemisphere mid-latitudes gradually decreases since the LGM, which matches well with reconstructed moisture index. Effective moisture change in most closed basins of the low-latitudes (monsoon regions) presents an opposite changing trend with that in the mid-latitudes. In the Asian mid-latitude closed basins, climate change in regions dominated by westerly winds exhibits a relative humid climate in the LGM and MH, and a relative dry climate in early and late Holocene. Whereas, East Asian summer monsoon generally influences the climate change in closed basins dominated by monsoons, which brings more water vapor over the early-to-mid Holocene but less water vapor in the LGM and late Holocene.

*Data Availability.* The TraCE-21kyr dataset comes from Climate Data Gateway at National Center for Atmospheric Research (NCAR) website https://www.earthsystemgrid.org/project/trace.html. PMIP3/CMIP5 simulations are available from the Earth System Grid Federation (ESGF) Peer-to-Peer (P2P) enterprise system website https://esgf-node.llnl.gov/projects/esgf-llnl/. Global closed basins boundaries are derived from the Hydrological data and maps based on SHuttle Elevation Derivatives at multiple Scales (HydroSHEDS) website https://www.hydrosheds.org/page/hydrobasins.

*Author contributions.* Yu Li and Yuxin Zhang designed this study and carried it out.

*Competing interests.* The authors declare that they have no conflict of interest.

*Acknowledgements.* This work was supported by the Second Tibetan Plateau Scientific Expedition and Research Program (STEP) (Grant No. 2019QZKK0202), the National Natural Science Foundation of China (Grant No. 42077415, 41822708), the Strategic Priority Research Program of Chinese Academy of Sciences (Grant No. XDA20100102).

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
