# Peer review of "Synergy of the westerly winds and monsoons in lake evolution of glo bal closed basins since the Last Glacial Maximum and its implicat ion for hydrological change in Central Asia"

_Climate of the Past, 2020_

## Referee Comment (RC1) · Anonymous Referee #1 · 21 Jun 2020

General comments The study presents an interesting way of separate the influence of westerlies and monsoon on mid-latitude closed basins by complementing paleoclimates records and climate models. However, minor changes should be made before final publication. 1. Most of the work and its conclusions are applicable to the Northern Hemisphere (NH); in my opinion this should be represented in the title of the work. 2. In Material and Methods section, authors consider three periods (LGM, MH and PI); however, in most of the analyses only LGM and Holocene are studied, having only a few mentions about the late Holocene or PI period. 3. I am little confused, in P7, L144 said "Whereas, effective moisture increases since the LGM over the global Tropics". However, one the main conclusions of this work is that monsoon areas were characterized by dry conditions during the LGM (and late Holocene), and humid conditions during the early-mid Holocene. Please could you explain that? 4. Figure 2: What are the dark areas in the map? Letters (a) and (b) are missing. 5. Figures 3 and 4: Improve figure caption, is not totally representative of the figure. 6. In P9, L177 text indicate that a moisture index was reconstructed from early Holocene to late Holocene. However, in methodology that fact is not totally explained.

Specific comments 1. Figure 3: For reduce unnecessary information on Figure 3, only include latitude at one side of the map 2. Figure caption on figure 6: I think that letters "(a)", "(b)" and "(c)" must go at the beginning of each description. 3. P4, L93, 96 and 100: It must be Equation (1) instead of Eq (2) 4. P4, L101: It must be Equation (2) instead of Eq. (3) 5. P13, L231: Text is confusing: "Major trend of moisture conditions revealed by the (. . .) is a gradual decrease since the early Holocene, and reaches the wettest status between 8 and 6 kyr in the East Asian monsoon region". It describes a decrease in moisture but ends with wettest conditions. Please reword the sentence in order to avoid confusion.

Technical corrections P2, L50: "Simulate" instead of "simulating". Include (precipitation minus evaporation) after P-E P2, L51: delete space before Pre-Industrial P2, L51-58: The sentence is too long, needs to be rephrased. P2, L53: "which" instead of "where" P2, L57: add a "s" at the end of monsoon (= monsoons) P2, L58: "Last" instead of "last" (capital letter) P2, L58-61: I think that the phrase "(..) according to records of Quaternary ice sheets, low-mid latitudes summer insolation and winter insolation, $\delta$18 O of Greenland ice core, etc." could be summarized.

P3, L67: delete space before 3 in CCSM3 P3, L69: delete space before 4 in CCSM4 P3, L84: Hostetler and Bartlein (1990)'s model

P4, L90: Add parenthesis to the referenced cited (= Morrill (2004) and Li and Morrill (2010)) P4, L94: Add a space after AB P4, L95: Add space before parenthesis "lake(m year-1)"

P4, L104: Add parenthesis to the referenced cited (= Li and Morrill (2010)). Replace Eq. (2) by (1) and (3) by (2). P4, L108: Delete "and" and replace phrase "and lake status information sorted by latitudes are shown in Table 2" by "Lake status information sorted by latitudes are shown in Table 2".

P6, Fig. 1: In figure caption replace "mm/year" by "mm year-1"

P7, L138: Replace "that lakes with" by "in which lakes with"or "where lakes with". It is not clear to me if Qinghai Lake, Hala Lake and Zhabuye are examples of lake with relative high-level during MH or PI. P7, L139: "and Zhabuye Lake..."

P9, Fig. 3: In figure caption add ", respectively." After "Wang et al. (2008)"

P10, L185: Add "that" before "a humid climate" P10, L186: Delete "And" at the beginning of the phrase P10, L187: Delete "the" before "paleoclimate modelling" P10, L188: Text is confusing, needs rewording "...resulting in the loss of lake water reduces and the high lake level sustains." P10, L190: replace "to increase" by "increasing"

P11, L201: include "and late Holocene" after "prevailed in the early Holocene".

Line 232: The phrase could be written as "The longest and highest-resolution drill core from Lake Qinghai (An et al. 2012) indicate that summer monsoon record generally (...)

P14, L255: Change sentence by "In these regions, winter precipitation accounts for a large proportion of annual precipitation (Li et al., 2008)".

---

## Author Comment (AC1) · 24 Jun 2020

We thanks very much for reviewer's comments concerning our manuscript. We use bold font to highlight reviewer's comments and use normal font to mark our reply.

**General comments The study presents an interesting way of separate the influence of westerlies and monsoon on mid-latitude closed basins by complementing paleoclimates records and climate models. However, minor changes should be made before final publication.**

**1. Most of the work and its conclusions are applicable to the Northern Hemisphere (NH); in my opinion this should be represented in the title of the work.**

Thank you very much for your suggestion. Our study regarded global closed basins as study area and explored synergy of the westerly winds and monsoons in lake evolution since the LGM. However, most closed basins are located in the Northern Hemisphere so that most of the work and conclusions are concentrated there. We will supplement the relevant conclusions of Southern Hemisphere in the revised version.

**2. In Material and Methods section, authors consider three periods (LGM, MH and PI); however, in most of the analyses only LGM and Holocene are studied, having only a few mentions about the late Holocene or PI period.**

Thank you very much for your suggestion. We will supplement the analyses of climate characteristic in the PI period in the revised version.

**3. I am little confused, in P7, L144 said "Whereas, effective moisture increases since the LGM over the global Tropics". However, one the main conclusions of this work is that monsoon areas were characterized by dry conditions during the LGM (and late Holocene), and humid conditions during the early-mid Holocene. Please could you explain that?**

Yes, of course. Based on the time series of the effective moisture change in the monsoon dominated closed basins

of the Northern Hemisphere, we draw a conclusion that humid climate prevails in the early-mid Holocene and relative dry climate appears in the LGM and late Holocene. However, according to the trend analysis of continuous simulation in effective moisture change, effective moisture increases since the LGM over the global Tropics. Trend analysis is used to judge whether the fluctuation of the time series is mainly rising or falling. Even effective moisture is relatively low in the LGM and late Holocene, and relatively high in the early-mid Holocene, the fluctuation of effective moisture is dominated by rising trend.

**4. Figure 2: What are the dark areas in the map? Letters (a) and (b) are missing.**

Thank you for pointing this out, the dark areas are global closed basins. And we will supplement the caption and Letters (a) and (b) of Figure 2 in the revised version.

[Figure]

**Figure 2.** (a) Distribution of global closed basins and circulation system: The dark areas are global closed basins; summer and winter are in accordance with the Northern Hemisphere; the shadows present the six monsoon areas according to Wang (2009), and (b) Trend analysis of continuous simulation in water balance change: The shadows

indicate that the trends are statistically significant at 5% level.

**5. Figures 3 and 4: Improve figure caption, is not totally representative of the figure.**

Thank you very much for your suggestion. We will improve figure caption of Figures 3 and 4.

**Figure 3.** (a) Spatial distribution feature of EOF1, (b) Spatial distribution feature of EOF2, (c) PCA1 and PCA2 of simulated water balance change since the LGM, and (d) Comparison between stalagmite records and summer insolation: Stalagmite records come from Dykoski et al. (2005) and Wang et al. (2008), summer insolation comes from Berger (1978).

**Figure 4.** Comparison between simulated water balance change and reconstructed moisture index in the mid-latitude closed basins of the Northern Hemisphere during the Holocene. Triangles indicate locations of paleoclimate records (Table 3).

**6. In P9, L177 text indicate that a moisture index was reconstructed from early Holocene to late Holocene. However, in methodology that fact is not totally explained.**

Thank you very much for your suggestion. We will clarify this in the revised version.

**Specific comments 1. Figure 3: For reduce unnecessary information on Figure 3, only include latitude at one side of the map.**

Thank you very much for your suggestion. We will modify the Figure 3 in the revised version.

[Figure]

**Figure 3.** (a) Spatial distribution feature of EOF1, (b) Spatial distribution feature of EOF2, (c) PCA1 and PCA2 of simulated water balance change since the LGM, and (d) Comparison between stalagmite records and summer insolation: Stalagmite records come from Dykoski et al. (2005) and Wang et al. (2008), summer insolation comes from Berger (1978).

**2. Figure caption on figure 6: I think that letters "(a)", "(b)" and "(c)" must go at the beginning of each description.**

Thank you very much for your suggestion. We will modify figure caption on Figure 6.

**Figure 6.** (a) Simulated water balance change between westerly dominated regions and monsoon regions in the Asian closed basin since the LGM, (b) General climate changing patterns during the Holocene in monsoon Asia and westerly Central Asia come from Chen et al. (2006), and (c) Extracted westerly dominated regions and monsoon regions in the Asian closed basins.

**3. P4, L93, 96 and 100: It must be Equation (1) instead of Eq (2).**

Thank you very much for your careful examination of the manuscript. We will modify this in the revised version.

**4. P4, L101: It must be Equation (2) instead of Eq. (3).**

Thank you very much for your careful examination of the manuscript. We will modify this in the revised version.

**5. P13, L231: Text is confusing: "Major trend of moisture conditions revealed by the (. . .) is a gradual decrease since the early Holocene, and reaches the wettest status between 8 and 6 kyr in the East Asian monsoon region". It describes a decrease in moisture but ends with wettest conditions. Please reword the sentence in order to avoid confusion.**

Thank you very much for your suggestion. We will reword this sentence.

**Technical corrections P2, L50: "Simulate" instead of "simulating". Include (precipitation minus evaporation) after P-E.**

Thank you very much for your careful examination of the manuscript. We will modify this.

**P2, L51: delete space before Pre-Industrial.**

Thank you very much for your careful examination of the manuscript. We will delete space before Pre-Industrial.

**P2, L51-58: The sentence is too long, needs to be rephrased.**

Thank you very much for your suggestion. We will reword this sentence.

**P2, L53: "which" instead of "where".**

Thank you very much for your careful examination of the manuscript. We will modify this.

**P2, L57: add a "s" at the end of monsoon (= monsoons).**

Thank you very much for your careful examination of the manuscript. We will modify this.

**P2, L58: "Last" instead of "last" (capital letter).**

Thank you very much for your careful examination of the manuscript. We will modify this.

**P2, L58-61: I think that the phrase "(..) according to records of Quaternary ice sheets, low-mid latitudes summer insolation and winter insolation, δ18 O of Greenland ice core, etc." could be summarized.**

Thank you very much for your suggestion. We will reword this sentence.

**P3, L67: delete space before 3 in CCSM3.**

Thank you very much for your careful examination of the manuscript. We will modify this.

**P3, L69: delete space before 4 in CCSM4.**

Thank you very much for your careful examination of the manuscript. We will modify this.

**P3, L84: Hostetler and Bartlein (1990)'s model.**

Thank you very much for your careful examination of the manuscript. We will modify this.

**P4, L90: Add parenthesis to the referenced cited (= Morrill (2004) and Li and Morrill (2010)).**

Thank you very much for your careful examination of the manuscript. We will modify this.

**P4, L94: Add a space after AB.**

Thank you very much for your careful examination of the manuscript. We will modify this.

**P4, L95: Add space before parenthesis "lake(m year-1)".**

Thank you very much for your careful examination of the manuscript. We will modify this.

**P4, L104: Add parenthesis to the referenced cited (= Li and Morrill (2010)). Replace Eq. (2) by (1) and (3) by**

**(2).**

Thank you very much for your careful examination of the manuscript. We will modify this.

**P4, L108: Delete "and" and replace phrase "and lake status information sorted by latitudes are shown in T able 2" by "Lake status information sorted by latitudes are shown in T able 2".**

Thank you very much for your careful examination of the manuscript. We will modify this.

**P6, Fig. 1: In figure caption replace "mm/year" by "mm year-1".**

Thank you very much for your careful examination of the manuscript. We will modify this.

**P7, L138: Replace "that lakes with" by "in which lakes with"or "where lakes with". It is not clear to me if Qinghai Lake, Hala Lake and Zhabuye are examples of lake with relative high-level during MH or PI.**
**P7, L139: "and Zhabuye Lake. . ."**

Thank you very much for your careful examination of the manuscript. We will reword this sentence.

**P9, Fig. 3: In figure caption add ", respectively." After "Wang et al. (2008)".**

Thank you very much for your careful examination of the manuscript. We will reword this figure caption.

**P10, L185: Add "that" before "a humid climate".**

Thank you very much for your careful examination of the manuscript. We will modify this.

**P10, L186: Delete "And" at the beginning of the phrase.**

Thank you very much for your careful examination of the manuscript. We will modify this.

**P10, L187: Delete "the" before "paleoclimate modelling".**

Thank you very much for your careful examination of the manuscript. We will modify this.

**P10, L188: Text is confusing, needs rewording ". . .resulting in the loss of lake water reduces and the high lake level sustains."**

Thank you very much for your suggestion. We will reword this sentence.

**P10, L190: replace "to increase" by "increasing".**

Thank you very much for your careful examination of the manuscript. We will modify this.

**P11, L201: include "and late Holocene" after "prevailed in the early Holocene". Line 232: The phrase could be written as "The longest and highest-resolution drill core from Lake Qinghai (An et al. 2012) indicate that summer monsoon record generally(. . .).**

Thank you very much for your suggestion. We will reword this sentence.

**P14, L255: Change sentence by "In these regions, winter precipitation accounts for a large proportion of annual precipitation (Li et al., 2008)".**

Thank you very much for your careful examination of the manuscript. We will modify this.

---

## Referee Comment (RC2) · Anonymous Referee #2 · 9 Aug 2020

General Comments The study combines simulated water balance in closed lake basins and paleoclimate records to distinguish the influence and temporal evolution of monsoon and mid-latitude westerlies on moisture levels. This study is an interesting approach to the influence of both the westerly winds and monsoon on climate changes since the Last Glacial Maximum. While as a whole the study is of good quality and fits within the scope of the journal, there are a number of issues with the manuscript, that I think will need to be taken care of prior to publication. - The authors present the study as global, but mainly focus on Central and East Asia. - Some changes in the structure of the manuscript are needed, especially in the results and discussion sections. - More details on the method of selection of the paleorecords is needed. - I think

the manuscript would greatly benefit from a thorough review of the English. While, the manuscript is comprehensible, there are many sentences that are not properly structured. The verb tense should be standardized, as they are sometimes changing even within a single sentence.

Specific comments Title I have issues with the title where the authors present the study as global, while in fact it is focusing on the Northern Hemisphere. The authors even provide the reasoning behind the focusing on the Northern Hemisphere in the last paragraph of the introduction. Actually, the study largely focuses on Central Asia and China (17/25 (68%) records from China). I think the title should be modified accordingly.

Introduction There is no clearly defined objective. Please clearly state the purpose of the study. What scientific question was this study intended to answer?

Time period partitioning What is the reasoning behind the selection of the PI period in the simulation? The authors mention that the selection of the time periods where subjective, was that 100 years period selected as a reference for the "modern/recent"? Why not choose a more climatically significant period like the Little Ice Age or the late Holocene, for which monsoon reconstruction clearly display a change? The authors mention that the division into those three periods was done to validate the water balance simulations and explore the evolution of the monsoons and westerly winds in the selected basins. Validating the water balance simulations for such a short period of time with records that are generally poorly constrained (see comment on section 2.2 below) for that period might be problematic. Furthermore, the PI period is absent from the discussion on the changes in monsoon and the westerlies.

Section 2.2: Please define what is considered a reliable chronology. . . Did the authors apply a minimum number of dates per thousand years? What a about the temporal resolution for the selection of the various records? Did the authors apply a minimum number of the samples per time frame? For example, minimum one sample per 100 or 200 years? I cannot tell for other regions, but to me there are some Chinese highresolution lake records missing from the list that would be of better quality than some of those included. On the top of my head, I would consider Gonghai lake (Chen et al., 2015 Sci Rep 5), Dali lake (Goldsmith et al., 2017 PNAS 114). They might not be within your simulated closed basins, but they are close enough and high-quality enough to be considered. Finally, for the PI period, as far as I know, many of the records in table 3 do not have any proper chronological control (210Pb or 14C bomb pulse) for the top section of the cores. The 1800-1900AD period can be difficult to narrow down chronologically as 14C is not very precise during this period and 210-Pb is at its limit.

In section 3.2, the authors state "Qinghai Lake, Hala Lake, Zhabuye Lake are typical lakes which are located in interactional transition zones between Asian monsoon and westerly winds, probably not following a single climate changing pattern". I would argue that many of the selected lakes in China, which they consider as being in the monsoon zones (see Fig. 6), were influenced both by the westerlies and the East Asian summer monsoon. Especially since the boundary of the monsoon was not static over time.

Structure of the manuscript Some parts of the result section belong to the discussion. While I understand that the authors must show that the lake simulations are valid and that, to do so, some interpretation is needed. I think that sections 3.3 and 3.4 should at the very least be moved to the discussion as they are focusing on the mechanisms driving the changes in water balance. Actually, I think that, given the nature of the data, this manuscript is a case where it would be beneficial to do a results and discussion section rather than separating them.

Terminology Several times in the manuscript, the authors refer to the Asian monsoon. To me it seems that what they call Asian monsoon is actually the East Asian monsoon. Especially since most of the selected records at the eastern edge of the simulated closed basins in Asia are roughly located at the northern limit of the East Asian summer monsoon (EASM). I think some precision is needed.

Discussion - Westerlies-monsoon interactions While studies have shown that trends in

moisture changes in Westerly dominated arid Central Asia generally differ from those in EASM regions, owing to the fact that EASM rainfall does not reach this region, the opposite is not necessarily true. Records well into the region that the authors would consider as the East Asian monsoon region suggest an influence of the westerlies on moisture levels. The authors briefly discuss the interactions between the westerlies and the East Asian monsoon. However, I think the discussion would benefit from a more in-depth discussion of the relationship between the Westerly Jet and the EASM. For example, there are increasing evidence for a control of the Westerly Jet on the northward extent and timing of the EASM rainfall in East Asia (see for example: Chiang et al., 2015 QSR 108: 11-129; Herzschuh et al., 2019 Nat Comm 10; Nagashima et al., 2013 (Geochem Geophys Geosys 14: 5041-5053).

- Speleothems The close similarity of the PCA1 time series with the speleothem records from Gongge and Hulu caves suggest it is a record of the East Asian summer monsoon. There is a long-standing debate about what the $\delta$18O speleothem records from China represents. One view interprets the oxygen isotopic record from Chinese cave deposits as reflecting real rainfall changes and hence reflecting changes in the EASM. The other main view suggests that these the oxygen records (depending where they are located) reflect changes in the moisture source (Indian monsoon vs EASM) and that they do not directly represent changes in EASM. What can the present study contribute to that debate? I think it could be an interesting addition to this manuscript.

Technical/minor comments Fig 3: Please provide letters to refer to each section of the figure both in the figure caption and the figure itself. I would also suggest putting both EOF figures on the left side and the PCA curves above the speleothem records. It would make the comparison of the curve easier. Fig 5: please provide letters the refer to each timeseries, especially since the font size is quite small. If possible, increase the font size of the time series. Section 3.3 and 3.4: EOF is not defined anywhere in the manuscript. Line 29: indicate rather than indicated. I would also remove monsoon after Australian and East African. Line 30: Remove the And at the start of the sentence.

Lines 32-33. That sentence needs to be rephrased to something like ". . . the seasonal migration of the (ITCZ) profoundly influences the seasonality of the global monsoons." Line 36: Please define LGM. This is the first time you mention it in the main body of the manuscript. Line 36: . . . southern regions of THE North American continent. . . Line 51: Please define MH. This is the first time you mention it in the main body of the manuscript. Line 51: remove one space between and and Pre-Industrial Line 51: remove and at the start of the sentence. Line 58: Capital letter for last Line 70: please define P-E. It is mentioned for the first time in the manuscript. Line 75: either remove And at the start of the sentence or combine with the previous one by for example writing: "(Peltier, 2004), while the vegetation. . ." Line 82: IN each grid cell not at Line 91: assumed rather than supposed Lines 135-136: However, there are exceptions that lakes. . . Replace that by where Lines 148-149: this sentence need to be rephrased, for example: " Comparing the simulations with the records, most simulations coincide with the upward. . ." Line 135 "For better validating simulated results, reviewed and summarized the millennial-scale changing patterns in lake level of the closed basins since the LGM are particularly important." Line 164: East Asian summer monsoon not East summer Asian. . . Line 173: suggested change to: "According to. . . basins in the Northern Hemisphere, affected both by low-latitude monsoon and mid-latitude westerly winds, are ideal region. . ." Line 179: "from A humid climate IN the early-mid Holocene to AN arid climate IN the late Holocene" not "from humid climate of the early-mid Holocene to arid climate of the late Holocene". Line 185: "in THE early-mid Holocene" Lines 187-188: That sentence need to be rewritten. Lines 188-190: Do you still refer to Yu et al. (2000) there or to Fig. 5. This is not clear. Line 190: reaches A maximum not the Line 221: experienced not experiences Line 255-256: suggest edit: "Winter precipitations account for a large proportion of annual precipitations in these regions."

---

## Author Comment (AC2) · 18 Aug 2020

We thanks very much for reviewer's comments concerning our manuscript. We use bold font to highlight reviewer's comments and use normal font to mark our reply.

**General Comments The study combines simulated water balance in closed lake basins and paleoclimate records to distinguish the influence and temporal evolution of monsoon and mid-latitude westerlies on moisture levels. This study is an interesting approach to the influence of both the westerly winds and monsoon on climate changes since the Last Glacial Maximum. While as a whole the study is of good quality and fits within the scope of the journal, there are a number of issues with the manuscript, that I think will need to be taken care of prior to publication.**

**1. The authors present the study as global, but mainly focus on Central and East Asia.**

Thank you very much for your suggestion. According to our original intention, we regard the global closed basins with prominent water resources problem as a carrier to explore the natural driving mechanisms that affect their dry and wet patterns—synergy of the global westerly winds and monsoons. However, most of the global closed basins are located in the Northern Hemisphere and the Eurasian continent has the largest area. Therefore, based on the differentiation of water balance change since the LGM in different latitudes of global closed basins, we focus on the mid-latitudes of the Northern Hemisphere where may be affected by the synergy of the westerly winds and monsoons to retrospect the water balance change since the LGM in the entire region. Then we further pay attention to the Eurasian continent for distinguishing the evolutionary characteristic of water balance in the regions respectively affected by the westerly winds and monsoons. Following your comments, we will revise the title to "Synergy of the westerly winds and monsoons in lake evolution of closed basins since the Last Glacial Maximum and its implication for hydrological change in Central Asia."

**2. Some changes in the structure of the manuscript are needed, especially in the results and discussion sections.**

Thank you very much for your suggestion. As you mentioned in Specific comments 6, we will combine sections 3.3 and 3.4 with the discussion to create a results and discussion section with a more readable description.

**3. More details on the method of selection of the paleorecords is needed.**

Thank you very much for your suggestion. As you mentioned in Specific comments 4, we will supplement the number of dating samples and resolution of paleoclimate records in Table 3.

**4. I think the manuscript would greatly benefit from a thorough review of the English. While, the manuscript is comprehensible, there are many sentences that are not properly structured. The verb tense should be standardized, as they are sometimes changing even within a single sentence.**

Thank you very much for your careful examination of the manuscript. We will check the each sentence carefully.

**Specific comments 1. Title I have issues with the title where the authors present the study as global, while in fact it is focusing on the Northern Hemisphere. The authors even provide the reasoning behind the focusing on the Northern Hemisphere in the last paragraph of the introduction. Actually, the study largely focuses on Central Asia and China (17/25 (68%) records from China). I think the title should be modified accordingly.**

Thank you very much for your suggestion. As answered in the General Comments 1, we will revise the title to "Synergy of the westerly winds and monsoons in lake evolution of closed basins since the Last Glacial Maximum and its implication for hydrological change in Central Asia."

**2. Introduction There is no clearly defined objective. Please clearly state the purpose of the study. What scientific question was this study intended to answer?**

Thank you very much for your suggestion. Global closed basins with prominent water resources problem occupy one-fifth of the terrestrial surface, distributing in both low-latitude monsoon regions and mid-latitude westerlies. As the two important components of global atmospheric circulation, monsoon system and westerly circulation exert different effects on global climate change and interact with each other in the mid-to-low latitudes. We regard the global closed basins as a carrier to explore the synergy of the westerly winds and monsoons in lake evolution since the LGM and its implication for hydrological change in Central Asia. We will modify the introduction for clearly stating the purpose of the study.

**3. Time period partitioning What is the reasoning behind the selection of the PI period in the simulation? The authors mention that the selection of the time periods where subjective, was that 100 years period**

**selected as a reference for the "modern/recent"? Why not choose a more climatically significant period like the Little Ice Age or the late Holocene, for which monsoon reconstruction clearly display a change? The authors mention that the division into those three periods was done to validate the water balance simulations and explore the evolution of the monsoons and westerly winds in the selected basins. Validating the water balance simulations for such a short period of time with records that are generally poorly constrained (see comment on section 2.2 below) for that period might be problematic. Furthermore, the PI period is absent from the discussion on the changes in monsoon and the westerlies.**

Thank you for pointing this out. Due to reduction of $CO_2$ levels at the LGM, previous studies hypothesize that lake levels of the LGM could map a reverse analog to future hydroclimate changes and verify this hypothesis by comparing hydroclimate change between the LGM and PI (Lowry and Morrill, 2019; Quade and Broecker, 2009). And in our study, PI period is a period with strong influence of modern human activities. In the time slice simulations, the selection of PI period is mainly used to measure the changes in hydroclimate conditions during the LGM and MH periods relative to the modern period, and verify the feasibility of the lake models by comparing the lake level simulation with the lake status records among the three periods. After verification, combining lake models and continuous simulation can be used to track water balance change of the global closed basins and investigate the evolutionary differentiation of the westerly winds and monsoons since the LGM. We agree that the absence of discussion on the changes of westerly winds and monsoons during the PI period is indeed an omission in our study, and we will add it accordingly in the revised version.

References:

Lowry, D. P. and Morrill, C. 2019. Is the Last Glacial Maximum a reverse analog for future hydroclimate changes in the Americas? Climate Dynamics, 52: 4407-4427.

Quade J, Broecker W S. 2009. Dryland hydrology in a warmer world: Lessons from the Last Glacial period. The European Physical Journal Special Topics, 176: 21-36.

**4. Section 2.2: Please define what is considered a reliable chronology. . . Did the authors apply a minimum number of dates per thousand years? What a about the temporal resolution for the selection of the various records? Did the authors apply a minimum number of the samples per time frame? For example, minimum one sample per 100 or 200 years? I cannot tell for other regions, but to me there are some Chinese high-resolution lake records missing from the list that would be of better quality than some of those included. On the top of my head, I would consider Gonghai lake (Chen et al., 2015 Sci Rep 5), Dali lake (Goldsmith et al., 2017 PNAS 114). They might not be within your simulated closed basins, but they are close enough and high-quality enough to be considered. Finally, for the PI period, as far as I know, many of the records in table 3 do not have any proper chronological control (210Pb or 14C bomb pulse)**

**for the top section of the cores. The 1800-1900AD period can be difficult to narrow down chronologically as 14C is not very precise during this period and 210-Pb is at its limit.**

Thank you very much for your suggestion. It is our negligence not to specify the number of dating samples and resolution of paleoclimate records in detail, and we will supplement these parts in Table 3 of the revised version. We fail to consider the paleoclimate records of Gonghai lake and Dali lake in our study, and as you suggested, these two high-resolution records will be added in the revised version. In this section, our aim is to reconstruct the regional moisture change by synthesizing the paleoclimate records for verifying the the continuous simulation. Therefore, we do not need to pay special attention to the dry and wet changes in the PI period, but focus on the matching degree of reconstructed results and simulated results throughout the Holocene. Both reconstructed moisture change and simulated water balance fluctuation exhibit a decreased trend since the early-Holocene, giving our confidence that the simulations are useful for investigating the evolutionary characteristics of the millennial westerly winds and monsoons.

**5. In section 3.2, the authors state "Qinghai Lake, Hala Lake, Zhabuye Lake are typical lakes which are located in interactional transition zones between Asian monsoon and westerly winds, probably not following a single climate changing pattern". I would argue that many of the selected lakes in China, which they consider as being in the monsoon zones (see Fig. 6), were influenced both by the westerlies and the East Asian summer monsoon. Especially since the boundary of the monsoon was not static over time.**

Indeed, due to various internal and external forces, the low-latitude monsoons and the mid-latitude westerly winds produce different intensities over time. The boundary of the East Asian summer monsoon will also be adjusted accordingly with the change of monsoon strength, leading to more complex and diverse evolution of Asian lakes. We will modify this sentence in the revised version as "Since the boundary of the monsoon will be adjusted accordingly with the change of East Asian summer monsoon strength, evolution of Asian lakes on the millennial scale probably not follows a single climate changing pattern."

**6. Structure of the manuscript Some parts of the result section belong to the discussion. While I understand that the authors must show that the lake simulations are valid and that, to do so, some interpretation is needed. I think that sections 3.3 and 3.4 should at the very least be moved to the discussion as they are focusing on the mechanisms driving the changes in water balance. Actually, I think that, given the nature of the data, this manuscript is a case where it would be beneficial to do a results and discussion section rather than separating them.**

Thank you very much for your suggestion. Your suggestion provides a new perspective for discussing our study

deeply. We will combine sections 3.3 and 3.4 with the discussion to create a results and discussion section with a more readable description.

**7. Terminology Several times in the manuscript, the authors refer to the Asian monsoon. To me it seems that what they call Asian monsoon is actually the East Asian monsoon. Especially since most of the selected records at the eastern edge of the simulated closed basins in Asia are roughly located at the northern limit of the East Asian summer monsoon (EASM). I think some precision is needed.**

Thank you very much for your suggestion. We will modify the "Asian monsoon" mentioned in this manuscript to the "East Asian summer monsoon" in the revised version.

**8. Discussion - Westerlies-monsoon interactions While studies have shown that trends in moisture changes in Westerly dominated arid Central Asia generally differ from those in EASM regions, owing to the fact that EASM rainfall does not reach this region, the opposite is not necessarily true. Records well into the region that the authors would consider as the East Asian monsoon region suggest an influence of the westerlies on moisture levels. The authors briefly discuss the interactions between the westerlies and the East Asian monsoon. However, I think the discussion would benefit from a more in-depth discussion of the relationship between the Westerly Jet and the EASM. For example, there are increasing evidence for a control of the Westerly Jet on the northward extent and timing of the EASM rainfall in East Asia (see for example: Chiang et al., 2015 QSR 108: 11-129; Herzschuh et al., 2019 Nat Comm 10; Nagashima et al., 2013 (Geochem Geophys Geosys 14: 5041-5053).**

Thank you very much for your suggestion. The information you provided about the influence of the orientation and position of the westerly jet on the EASM rainfall give us a lot of help. Previous studies mostly focus on the complexity of climate change in the transition zone between the westerlies and Asian monsoon, and investigate the interplay of two global atmospheric circulation on the millennial scale. However, the impact of the seasonal progression of the westerly jet on the EASM rainfall has not been thoroughly discussed. We will supplement this issue in the results and discussion section.

**9. Speleothems The close similarity of the PCA1 time series with the speleothem records from Gongge and Hulu caves suggest it is a record of the East Asian summer monsoon. There is a long-standing debate about what the δ18O speleothem records from China represents. One view interprets the oxygen isotopic**

**record from Chinese cave deposits as reflecting real rainfall changes and hence reflecting changes in the EASM. The other main view suggests that these the oxygen records (depending where they are located) reflect changes in the moisture source (Indian monsoon vs EASM) and that they do not directly represent changes in EASM. What can the present study contribute to that debate? I think it could be an interesting addition to this manuscript.**

Thank you very much for your suggestion. The climatic significance of the $\delta^{18}O_c$ in the Asian speleothem records is always a long-standing debate, and some influential hypotheses regard $\delta^{18}O_c$ of the monsoon regions as a proxy for "Asian monsoon intensity", "Indian monsoon intensity", "summer monsoon rainfall amount", "circulation conditions", etc. Although the climatic significance is controversial, it is well-accepted that $\delta^{18}O_c$ changes should bear the imprint of variations in the oxygen isotopic composition of precipitation ($\delta^{18}O_p$) (Cheng et al., 2012; Chen et al., 2016). In addition, a conclusion that the evolution of the EASM is generally controlled by summer insolation in the Northern Hemisphere is also widely recognized (Yuan et al., 2004; Chen et al., 2006; An et al., 2015). We therefore not only compare the PCA1 with the stalagmite records of Dongge Cave with controversial climatic significance, but also with the summer solar radiation at low-latitudes in the Northern Hemisphere. However, the contribution of our results to the paleoclimate research of Chinese stalagmites is not discussed in-depth in the article, and we will make corresponding supplement in the results and discussion section.

References:

Cheng H, Sinha A, Wang X, Cruz F W, Edwards R L. 2012. The global paleomonsoon as seen through speleothem records from Asia and the Americas. Climate Dynamics, 39: 1045-1062.

Chen J H, Rao Z G, Liu J B, Huang W, Feng S, Dong G H, Hu Y, Xu Q H, Chen F H. 2016. On the timing of the East Asian summer monsoon maximum during the Holocene—Does the speleothem oxygen isotope record reflect monsoon rainfall variability? Science China Earth Sciences, 59: 2328-2338.

Yuan D, Cheng H Y, Edwards R L, Dykoski C A, Kelly M J, Zhang M. 2004. Timing, Duration, and Transitions of the Last Interglacial Asian Monsoon. Science, 304: 575-578.

Chen F H, Huang X Z, Yang M L, Yang X L, Fan Y X, Zhao H. 2006. Westerly dominated Holocene climate model in arid central Asia—Case study on Bosten lake, Xinjiang, China. Quaternary Sciences, 26: 881-887.

An Z S, Wu G X, Li J P, Sun Y B, Liu Y M, Zhou W J, Cai Y J, Duan A M, Li L, Mao J Y, Cheng H, Shi Z G, Tan L C, Yan H, Ao H, Chang H, Feng J. 2015 Global Monsoon Dynamics and Climate Change. Annual Review of Earth and Planetary Sciences, 43: 2.1-2.49.

**Technical/minor comments Fig 3: Please provide letters to refer to each section of the figure both in the figure caption and the figure itself. I would also suggest putting both EOF figures on the left side and the PCA curves above the speleothem records. It would make the comparison of the curve easier.**

Thank you very much for your suggestion. We will modify the Figure 3 in the revised version.

**Fig 5: please provide letters the refer to each time series, especially since the font size is quite small. If possible, increase the font size of the time series.**

Thank you for pointing this out, we will provide letters the refer to each time series and increase the font size of the time series.

**Section 3.3 and 3.4: EOF is not defined anywhere in the manuscript.**

Thank you for pointing this out, we will add section 2.3 to describe the mathematical methods.

**Line 29: indicate rather than indicated. I would also remove monsoon after Australian and East African.**

Thank you very much for your careful examination of the manuscript. We will modify this.

**Line 30: Remove the And at the start of the sentence.**

Thank you very much for your careful examination of the manuscript. We will modify this.

**Lines 32-33. That sentence needs to be rephrased to something like ". . . the seasonal migration of the (ITCZ) profoundly influences the seasonality of the global monsoons."**

Thank you very much for your suggestion. We will rephrase this sentence in the revised version.

**Line 36: Please define LGM. This is the first time you mention it in the main body of the manuscript.**

Thank you very much for your careful examination of the manuscript. We will add the full name of LGM.

**Line 36: . . . southern regions of THE North American continent. . .**

Thank you very much for your careful examination of the manuscript. We will modify this.

**Line 51: Please define MH. This is the first time you mention it in the main body of the manuscript.**

Thank you very much for your careful examination of the manuscript. We will add the full name of MH.

**Line 51: remove one space between and and Pre-Industrial**

Thank you very much for your careful examination of the manuscript. We will modify this.

**Line 51: remove and at the start of the sentence.**

Thank you very much for your suggestion. We will modify this.

**Line 58: Capital letter for last**

Thank you very much for your careful examination of the manuscript. We will modify this.

**Line 70: please define P-E. It is mentioned for the first time in the manuscript.**

Thank you very much for your careful examination of the manuscript. We will add the full name of P-E.

**Line 75: either remove And at the start of the sentence or combine with the previous one by for example writing: "(Peltier, 2004), while the vegetation. . ."**

Thank you very much for your suggestion. We will modify this.

**Line 82: IN each grid cell not at**

Thank you very much for your suggestion. We will modify this.

**Line91: assumed rather than supposed**

Thank you very much for your suggestion. We will modify this.

**Lines 135-136: However, there are exceptions that lakes. . . Replace that by where**

Thank you very much for your suggestion. We will modify this.

**Lines 148-149: this sentence need to be rephrased, for example: " Comparing the simulations with the records, most simulations coincide with the upward. . ."**

Thank you very much for your suggestion. We will modify this.

**Line 135 "For better validating simulated results, reviewed and summarized the millennial-scale changing patterns in lake level of the closed basins since the LGM are particularly important."**

Thank you very much for your careful examination of the manuscript. We will modify this.

**Line 164: East Asian summer monsoon not East summer Asian. . .**

Thank you very much for your careful examination of the manuscript. We will modify this.

**Line 173: suggested change to: "According to. . . basins in the Northern Hemisphere, affected both by low-latitude monsoon and mid-latitude westerly winds, are ideal region. . ."**

Thank you very much for your suggestion. We will modify this.

**Line 179: "from A humid climate IN the early-mid Holocene to AN arid climate IN the late Holocene" not "from humid climate of the early-mid Holocene to arid climate of the late Holocene".**

Thank you very much for your careful examination of the manuscript. We will modify this.

**Line 185: "in THE early-mid Holocene"**

Thank you very much for your careful examination of the manuscript. We will modify this.

**Lines 187-188: That sentence need to be rewritten.**

Thank you very much for your suggestion. We will reword this sentence.

**Lines 188-190: Do you still refer to Yu et al. (2000) there or to Fig. 5. This is not clear.**

Thank you for pointing this out, lines 188-190 describe Fig. 5 and we will clarify it.

**Line 190: reaches A maximum not the**

Thank you very much for your suggestion. We will modify this.

**Line 221: experienced not experiences**

Thank you very much for your suggestion. We will modify this.

**Line 255-256: suggest edit: "Winter precipitations account for a large proportion of annual precipitations in these regions."**

Thank you very much for your suggestion. We will reword this sentence.

---

## Author Response (AR1)

Dear editor,

Thank you for your letter and for the reviewer's comments concerning our manuscript entitled "Synergy of the westerly winds and monsoons in lake evolution of global closed basins since the Last Glacial Maximum and its implication for hydrological change in Central Asia" (Manuscript No.: cp-2020-53). Thanks very much for your hard work to this paper. Those comments and suggestions are all valuable and very helpful for revising and improving our paper, as well as the important guiding significance to our researches. We appreciated the comments and suggestions very much. According to the comments from the reviewers, we completely revised the paper. Particularly, we highlighted the study area and supplemented the relevant results of PI period or the late Holocene, as well as reworked the structure of the "results and discussion" sections in the revised version. A point-by-point reply to the comments, a list of all relevant changes made in the manuscript, and a marked-up manuscript version are listed below this letter. We look forward to your satisfaction with this revision.

Yours sincerely,

Yu Li

Corresponding author:

Name: Yu Li

Address: College of Earth and Environmental Sciences, Lanzhou University, Lanzhou 730000, China

E-mail: liyu@lzu.edu.cn

**A point-by-point response to the reviews**

**Response to Referee #1**

**General responses:**

**Reviewer 1 concerned that the title of the article does not highlight the main research area and climate change in the late Holocene or PI period has been less discussed. In the revised version, we modified the title of this article and supplemented relevant contents about climate change in the late Holocene or PI period. Also, she/he has provided some specific comments on the manuscript, and we revised the manuscript as he/she suggests. A detailed list of changes against each point which is being raised was in below.**

**General comments The study presents an interesting way of separate the influence of westerlies and monsoon on**

mid-latitude closed basins by complementing paleoclimates records and climate models. However, minor changes should be made before final publication.

**1. Most of the work and its conclusions are applicable to the Northern Hemisphere (NH); in my opinion this should be represented in the title of the work.**

Response: Thank you very much for your suggestion. Our study regarded global closed basins with prominent water resources problem and explored synergy of the westerly winds and monsoons in lake evolution since the LGM. We first discovered that there is a significant differentiation between the monsoon regions and the westerlies in the evolution of water balance over the global closed basins, especially the Northern Hemisphere. Then we focused on the climate change in closed basins of the Northern Hemisphere which is affected both by low-latitude monsoons and mid-latitude westerly winds. Further, we emphatically investigated the millennial scale evolution characteristics and mechanisms of East Asian summer monsoon and westerly winds in closed basins of the Asian continent since the LGM. Following your comments, we revised the title to "Synergy of the westerly winds and monsoons in lake evolution of global closed basins since the Last Glacial Maximum and its implication for hydrological change in Central Asia" in the revised version.

**2. In Material and Methods section, authors consider three periods (LGM, MH and PI); however, in most of the analyses only LGM and Holocene are studied, having only a few mentions about the late Holocene or PI period.**

Response: Thank you very much for your suggestion. We supplemented the analyses of climate characteristics in the PI period or the late Holocene in the revised version.

**3. I am little confused, in P7, L144 said "Whereas, effective moisture increases since the LGM over the global Tropics". However, one the main conclusions of this work is that monsoon areas were characterized by dry conditions during the LGM (and late Holocene), and humid conditions during the early-mid Holocene. Please could you explain that?**

Response: Yes, of course. Based on the time series of the effective moisture change in the monsoon dominated closed basins of the Northern Hemisphere, we draw a conclusion that humid climate prevails in the early-to-mid Holocene and relative dry climate appears in the LGM and late Holocene. However, according to the trend analysis of continuous simulation in

effective moisture change, effective moisture increases since the LGM over the global Tropics. Even effective moisture is relatively low in the LGM and late Holocene, and relatively high in the early-to-mid Holocene, the fluctuation of effective moisture is dominated by rising trend. Trend analysis is used to judge whether the fluctuation of the time series is mainly rising or falling, and we added this mathematical method in section 2.3. The updated contents are reproduced below.

"**2.3 Mathematical methods**

Linear tendency estimation is a common trend analysis method, which was chosen to measure the variation degree of simulated water balance in this paper. Besides, we also used the Empirical orthogonal function (EOF), a method of analyzing the structural features in matrix data and extracting the feature vector of main data, to examine spatially and temporally variability of simulated water balance. The spatial distribution of EOF first (second) mode is denoted by EOF1 (EOF2), and the time series of first (second) mode is denoted by PCA1 (PCA2)."

**4. Figure 2: What are the dark areas in the map? Letters (a) and (b) are missing.**

Response: Thank you for pointing this out, the dark areas are global closed basins. And we supplemented the caption and letters (a) and (b) of Figure 2 in the revised version. The updated contents are reproduced below.

[Figure]

"**Figure 2.** (a) Distribution of global closed basins and circulation system: The dark areas are global closed basins; summer and winter of the ITCZ are in accordance with the Northern Hemisphere; the shadows present the six monsoon areas according to Wang (2009), and (b) Trend analysis of continuous simulation in water balance change: The shadows indicate that the trends are statistically significant at 5% level."

**5. Figures 3 and 4: Improve figure caption, is not totally representative of the figure.**

Response: Thank you very much for your suggestion. We improved figure caption of Figures 3 and 4. The updated contents are reproduced below.

"**Figure 3.** (a) Spatial distribution feature of EOF1, (b) PCA1 and PCA2 of simulated water balance change since the LGM, (c) Spatial distribution feature of EOF2, and (d) Comparison between stalagmite records and summer insolation: Stalagmite

records come from Dykoski et al. (2005) and Wang et al. (2008), and summer insolation comes from Berger (1978).

**Figure 4.** Comparison between simulated water balance change and reconstructed moisture index in the mid-latitude closed basins of the Northern Hemisphere during the Holocene. Triangles indicate locations of paleoclimate records (Table 3)."

**6. In P9, L177 text indicate that a moisture index was reconstructed from early Holocene to late Holocene. However, in methodology that fact is not totally explained.**

Response: Thank you very much for your suggestion. We explained this in the revised version. The updated contents are reproduced below.

"Due to the different time scales of the collected continuous paleoclimate records, we can only reconstruct the regional moisture change from the early to late Holocene after unifying the time scales, but the purpose of this part is only to check the simulation results."

**Specific comments 1. Figure 3: For reduce unnecessary information on Figure 3, only include latitude at one side of the map.**

Response: Thank you very much for your suggestion. We modified the Figure 3 in the revised version. The updated contents are reproduced below.

[Figure]

"**Figure 3.** (a) Spatial distribution feature of EOF1, (b) PCA1 and PCA2 of simulated water balance change since the LGM, (c) Spatial distribution feature of EOF2, and (d) Comparison between stalagmite records and summer insolation: Stalagmite records come from Dykoski et al. (2005) and Wang et al. (2008), and summer insolation comes from Berger (1978)."

**2. Figure caption on figure 6: I think that letters "(a)", "(b)" and "(c)" must go at the beginning of each description.**

Response: Thank you very much for your suggestion. We modified figure caption on Figure 6. The updated contents are reproduced below.

"**Figure 6.** (a) Simulated water balance change between westerly dominated regions and monsoon regions in the Asian closed basins since the LGM, (b) General climate changing patterns during the Holocene in monsoon Asia and westerly Central Asia come from Chen et al. (2006), and (c) Extracted westerly dominated regions and monsoon regions in the Asian closed basins."

**3. P4, L93, 96 and 100: It must be Equation (1) instead of Eq (2).**

Response: Thank you very much for your careful examination of the manuscript. We modified this in the revised version.

**4. P4, L101: It must be Equation (2) instead of Eq. (3).**

Response: Thank you very much for your careful examination of the manuscript. We modified this in the revised version.

**5. P13, L231: Text is confusing: "Major trend of moisture conditions revealed by the (. . .) is a gradual decrease since the early Holocene, and reaches the wettest status between 8 and 6 kyr in the East Asian monsoon region". It describes a decrease in moisture but ends with wettest conditions. Please reword the sentence in order to avoid confusion.**

Response: Thank you very much for your suggestion. We reworded this sentence. The updated contents are reproduced below.

"By comprehensively analyzing a variety of paleoclimate proxies, Wang et al. (2017) suggested that moisture change revealed by the Australian monsoon, the East African monsoon and the Indian monsoon regions reaches the wettest status in the early Holocene, while the wettest condition in the East Asian summer monsoon regions occurs between 8 and 6 kyr."

**Technical corrections P2, L50: "Simulate" instead of "simulating". Include (precipitation minus evaporation) after P-E.**

Response: Thank you very much for your careful examination of the manuscript. We modified this.

**P2, L51: delete space before Pre-Industrial.**

Response: Thank you very much for your careful examination of the manuscript. We deleted space before Pre-Industrial.

**P2, L51-58: The sentence is too long, needs to be rephrased.**

Response: Thank you very much for your suggestion. We reworded this sentence. The updated contents are reproduced below.

"The prominent spatial differentiation of monsoons and westerly winds revealed by simulations leads us to focus on the Northern Hemisphere mid-latitude closed basins which are simultaneously influenced by mid-latitude westerly winds and

low-latitude monsoons. In the mid-latitude closed basins of the Northern Hemisphere, the good match between water balance simulation and reconstructed moisture index from 27 paleoclimate records verifies the reliability of the simulation results. Further, we disaggregated the Northern Hemisphere mid-latitude closed basins into the areas dominated by monsoons and westerly winds respectively, and emphatically explored the temporal evolution of the East Asian summer monsoon and westerly winds since the LGM. According to the climate records, we comprehensively considered the determinants that control the trend of climate change in the Northern Hemisphere westerlies and East Asian summer monsoon regions since the LGM."

**P2, L53: "which" instead of "where".**

Response: Thank you very much for your careful examination of the manuscript. We modified this.

**P2, L57: add a "s" at the end of monsoon (= monsoons).**

Response: Thank you very much for your careful examination of the manuscript. We modified this.

**P2, L58: "Last" instead of "last" (capital letter).**

Response: Thank you very much for your careful examination of the manuscript. We deleted this.

**P2, L58-61: I think that the phrase "(..) according to records of Quaternary ice sheets, low-mid latitudes summer insolation and winter insolation, δ18 O of Greenland ice core, etc." could be summarized.**

Response: Thank you very much for your suggestion. We reworded this sentence. The updated contents are reproduced below.

"According to the climate records, we comprehensively considered the determinants that control the trend of climate change in the Northern Hemisphere westerlies and East Asian summer monsoon regions since the LGM."

**P3, L67: delete space before 3 in CCSM3.**

Response: Thank you very much for your careful examination of the manuscript. We modified this.

**P3, L69: delete space before 4 in CCSM4.**

Response: Thank you very much for your careful examination of the manuscript. We modified this.

**P3, L84: Hostetler and Bartlein (1990)'s model.**

Response: Thank you very much for your careful examination of the manuscript. We modified this.

**P4, L90: Add parenthesis to the referenced cited (= Morrill (2004) and Li and Morrill (2010)).**

Response: Thank you very much for your careful examination of the manuscript. We modified this.

**P4, L94: Add a space after AB.**

Response: Thank you very much for your careful examination of the manuscript. We modified this.

**P4, L95: Add space before parenthesis "lake(m year-1)".**

Response: Thank you very much for your careful examination of the manuscript. We modified this.

**P4, L104: Add parenthesis to the referenced cited (= Li and Morrill (2010)). Replace Eq. (2) by (1) and (3) by (2).**

Response: Thank you very much for your careful examination of the manuscript. We modified this.

**P4, L108: Delete "and" and replace phrase "and lake status information sorted by latitudes are shown in Table 2" by "Lake status information sorted by latitudes are shown in Table 2".**

Response: Thank you very much for your careful examination of the manuscript. We modified this. The updated contents are reproduced below.

"Lake status information sorted by latitudes is shown in Table 2."

**P6, Fig. 1: In figure caption replace "mm/year" by "mm year-1".**

Response: Thank you very much for your careful examination of the manuscript. We modified this.

**P7, L138: Replace "that lakes with" by "in which lakes with"or "where lakes with". It is not clear to me if Qinghai Lake, Hala Lake and Zhabuye are examples of lake with relative high-level during MH or PI.**

**P7, L139: "and Zhabuye Lake. . ."**

Response: Thank you very much for your careful examination of the manuscript. We deleted this sentence.

**P9, Fig. 3: In figure caption add ", respectively." After "Wang et al. (2008)".**

Response: Thank you very much for your careful examination of the manuscript. We reworded this figure caption. The updated contents are reproduced below.

"**Figure 3.** (a) Spatial distribution feature of EOF1, (b) PCA1 and PCA2 of simulated water balance change since the LGM, (c) Spatial distribution feature of EOF2, and (d) Comparison between stalagmite records and summer insolation: Stalagmite records come from Dykoski et al. (2005) and Wang et al. (2008), and summer insolation comes from Berger (1978)."

**P10, L185: Add "that" before "a humid climate".**

Response: Thank you very much for your careful examination of the manuscript. We modified this.

**P10, L186: Delete "And" at the beginning of the phrase.**

Response: Thank you very much for your careful examination of the manuscript. We modified this.

**P10, L187: Delete "the" before "paleoclimate modelling".**

Response: Thank you very much for your careful examination of the manuscript. We modified this.

**P10, L188: Text is confusing, needs rewording ". . .resulting in the loss of lake water reduces and the high lake level sustains."**

Response: Thank you very much for your suggestion. We reworded this sentence. The updated contents are reproduced

below.

"Using paleoclimate modelling, Yu et al. (2000) mentioned that the low temperature during the glacial period causes a decrease of evaporation and a reduction of lake water loss, resulting in the appearance of high lake level."

**P10, L190: replace "to increase" by "increasing".**

Response: Thank you very much for your careful examination of the manuscript. We modified this.

**P11, L201: include "and late Holocene" after "prevailed in the early Holocene".**

Response: Thank you very much for your suggestion. We modified this.

**Line 232: The phrase could be written as "The longest and highest-resolution drill core from Lake Qinghai (An et al. 2012) indicate that summer monsoon record generally(. . .).**

Response: Thank you very much for your suggestion. We reworded this sentence. The updated contents are reproduced below.

"The longest and highest-resolution drill core from Lake Qinghai (An et al., 2012) indicates that the summer monsoon record generally resembles the changing trends of Asian summer monsoon records derived from Dongge and Hulu speleothems over the last 20 kyr, and the mid-latitude westerlies climate dominates the Lake Qinghai area in glacial times."

**P14, L255: Change sentence by "In these regions, winter precipitation accounts for a large proportion of annual precipitation (Li et al., 2008)".**

Response: Thank you very much for your careful examination of the manuscript. We modified this. The updated contents are reproduced below.

"Winter precipitation accounts for a large proportion of annual precipitations in these regions (Li et al., 2008)."

**Response to Referee #2**

**General responses:**

**Reviewer 2 focused on the the title of the article and details of paleoclimate records, as well as the structure of the results and discussion sections. In the revised version, we modified the title of this article, supplemented details of paleoclimate records, reworked the results and discussion sections and added some new contents in different sections of this paper. Also, she/he has provided some specific comments on the manuscript, and we revised the manuscript as he/she suggests. A detailed list of changes against each point which is being raised was in below.**

**General comments The study combines simulated water balance in closed lake basins and paleoclimate records to distinguish the influence and temporal evolution of monsoon and mid-latitude westerlies on moisture levels. This study is an interesting approach to the influence of both the westerly winds and monsoon on climate changes since the Last Glacial Maximum. While as a whole the study is of good quality and fits within the scope of the journal, there are a number of issues with the manuscript, that I think will need to be taken care of prior to publication.**

**1. The authors present the study as global, but mainly focus on Central and East Asia.**

Response: Thank you very much for your suggestion. Our study regarded global closed basins with prominent water resources problem and explored synergy of the westerly winds and monsoons in lake evolution since the LGM. We first discovered that there is a significant differentiation between the monsoon regions and the westerlies in the evolution of water balance over the global closed basins, especially the Northern Hemisphere. Then we focused on the climate change in closed basins of the Northern Hemisphere which is affected both by low-latitude monsoons and mid-latitude westerly winds. Further, we emphatically investigated the millennial scale evolution characteristics and mechanisms of East Asian summer monsoon and westerly winds in closed basins of the Asian continent since the LGM. We made some detailed interpretations in the revised version.

**2. Some changes in the structure of the manuscript are needed, especially in the results and discussion sections.**

Response: Thank you very much for your suggestion. As you mentioned in Specific comments 6, we combined the original result sections with the original discussion sections to create a "results and discussion" section with four parts.

**3. More details on the method of selection of the paleorecords is needed.**

Response: Thank you very much for your suggestion. As you mentioned in Specific comments 4, we supplemented the number of dating samples and resolution of paleoclimate records in Table 3.

**4. I think the manuscript would greatly benefit from a thorough review of the English. While, the manuscript is comprehensible, there are many sentences that are not properly structured. The verb tense should be standardized, as they are sometimes changing even within a single sentence.**

Response: Thank you very much for your careful examination of the manuscript. We checked each sentence carefully.

**Specific comments 1. Title I have issues with the title where the authors present the study as global, while in fact it is focusing on the Northern Hemisphere. The authors even provide the reasoning behind the focusing on the Northern Hemisphere in the last paragraph of the introduction. Actually, the study largely focuses on Central Asia and China (17/25 (68%) records from China). I think the title should be modified accordingly.**

Response: Thank you very much for your suggestion. Our study regarded global closed basins with prominent water resources problem and explored synergy of the westerly winds and monsoons in lake evolution since the LGM. We first discovered that there is a significant differentiation between the monsoon regions and the westerlies in the evolution of water balance over the global closed basins, especially the Northern Hemisphere. Then we focused on the climate change in closed basins of the Northern Hemisphere which is affected both by low-latitude monsoons and mid-latitude westerly winds. Further, we emphatically investigated the millennial scale evolution characteristics and mechanisms of East Asian summer monsoon and westerly winds in closed basins of the Asian continent since the LGM. Following your comments, we revised the title to "Synergy of the westerly winds and monsoons in lake evolution of global closed basins since the Last Glacial Maximum and its implication for hydrological change in Central Asia" in the revised version.

**2. Introduction There is no clearly defined objective. Please clearly state the purpose of the study. What scientific question was this study intended to answer?**

Response: Thank you very much for your suggestion. We modified the introduction for clearly stating the purpose of the study. The updated contents are reproduced below.

"As important components of atmospheric circulation systems, the mid-latitude westerly winds and low-latitude monsoon systems play key roles in global climate change. Whether on the decadal or the millennial scale, researches about this aspect always attract widespread attention from scientists. Examination of global monsoon precipitation changes in land suggests an overall weakening over the recent half-century (1950-2000) (Zhou et al., 2008). Individual monsoon indexes reconstructed by Wang et al. (2017) indicate the moisture in the tropical Australian, the East Africa, and the Indian monsoon regions exhibits a gradual decrease since the early Holocene. It is widely accepted that the East Asian summer monsoon usually follows the variation of low-latitude summer solar radiation (Yuan et al., 2004; Chen et al., 2006; An et al., 2015). Charney (1969) and Wang (2009) also proposed that the seasonal migration of the intertropical convergence zone (ITCZ) profoundly influences the seasonality of the global monsoons. However, the global westerly winds and their associated storm tracks dominate the mid-latitude dynamics of the global atmosphere and affect the extratropical large-scale temperature and precipitation patterns (Oster et al., 2015; Voigt et al., 2015). Since the Last Glacial Maximum (LGM), climate in central and southern regions of the North American continent gradually dries out as the ice sheet melt and the westerlies move to north (Qin et al., 1997). As mentioned in the foregoing studies, millennial scale evolution in global monsoons and westerly winds probably shows different patterns as a result of complex driving mechanisms. Arguments about an asynchronous pattern of moisture variations between monsoon and westerly winds evolution underscore the importance of studying their millennial scale differentiation (Chen et al., 2006, 2008, 2019; An and Chen, 2009; Li et al., 2011; An et al., 2012).

A way to examine past climate variability is traditional methods of studying various archives which truly document the evolution of regional climate, including lake sediments (Madsen et al., 2008), stalagmites (Dykoski et al., 2005; Wang et al., 2008) and tree rings (Linderholm and Braeuning, 2006). However, due to the limited time scale of paleoclimate records, most researches on the evolution of monsoons and westerly winds are concentrated in the Holocene and lack an exploration during the LGM. With the development of paleoclimatology in recent decades, numerical simulations of paleoclimate continue to emerge and develop to a relatively mature system, which provides a useful tool for reviewing paleoclimate change over long time scales. On account of water balance system constantly responding to climatic conditions changes, a combination of numerical simulations and lake water balance models can be used to effectively track past climate change,

and make up the deficiency in qualitative method of multi-proxy analysis (Qin and Yu, 1998; Xue and Yu, 2000; Morrill et al., 2001, 2004; Li and Morrill, 2010, 2013; Lowry and Morrill, 2019; Li et al., 2020). Covering one-fifth of the terrestrial surface, global closed basins distribute in both low-latitude monsoon regions and mid-latitude westerlies. Furthermore, closed basins with relative independent hydrological cycle system have a plenty of terminal lakes records that provide more evidence for retrospecting climate change (Li et al., 2017), and can be regarded as ideal regions for studying spatiotemporal difference between monsoons and westerly winds."

**3. Time period partitioning What is the reasoning behind the selection of the PI period in the simulation? The authors mention that the selection of the time periods where subjective, was that 100 years period selected as a reference for the "modern/recent"? Why not choose a more climatically significant period like the Little Ice Age or the late Holocene, for which monsoon reconstruction clearly display a change? The authors mention that the division into those three periods was done to validate the water balance simulations and explore the evolution of the monsoons and westerly winds in the selected basins. Validating the water balance simulations for such a short period of time with records that are generally poorly constrained (see comment on section 2.2 below) for that period might be problematic. Furthermore, the PI period is absent from the discussion on the changes in monsoon and the westerlies.**

Response: Thank you for pointing this out. In the time slice simulations, the selection of PI period which is considered as a typical period of the late Holocene, is mainly used to measure the changes in hydroclimate conditions during the LGM and MH periods relative to the late Holocene, and verify the feasibility of the lake models by comparing the lake level simulation with the lake status records among the three periods. After verification, combining lake models and continuous simulation can be used to track water balance change of the global closed basins and investigate the evolutionary differentiation of the westerly winds and monsoons since the LGM. We added relevant explanations in the revised version and supplemented the discussion on the changes in monsoon and the westerlies during the late Holocene. The updated contents are reproduced below.

"Here the PI period which is considered as a typical period of the late Holocene, is mainly used to measure the changes of hydroclimate conditions during the LGM and MH periods relative to the late Holocene, and verify the feasibility of the lake models by comparing the lake level simulations with the lake status records among three periods."

"The regions dominated by East Asian summer monsoon and westerly winds were then selected respectively based on the spatial characteristics of two modes extracted from the EOF, to explore millennial scale evolution features of two climate systems (Fig. 6). In the westerly winds dominated regions, the LGM and MH are characterized by humid climate, and relative dry climate prevails in the early and late Holocene. Whereas, the water balance in the monsoon dominated regions is generally affected by East Asian summer monsoon which brings much water vapor over the early-to-mid Holocene, and leads to relative dry climate in the LGM and late Holocene."

**4. Section 2.2: Please define what is considered a reliable chronology. . . Did the authors apply a minimum number of dates per thousand years? What a about the temporal resolution for the selection of the various records? Did the authors apply a minimum number of the samples per time frame? For example, minimum one sample per 100 or 200 years? I cannot tell for other regions, but to me there are some Chinese high-resolution lake records missing from the list that would be of better quality than some of those included. On the top of my head, I would consider Gonghai lake (Chen et al., 2015 Sci Rep 5), Dali lake (Goldsmith et al., 2017 PNAS 114). They might not be within your simulated closed basins, but they are close enough and high-quality enough to be considered. Finally, for the PI period, as far as I know, many of the records in table 3 do not have any proper chronological control (210Pb or 14C bomb pulse) for the top section of the cores. The 1800-1900AD period can be difficult to narrow down chronologically as 14C is not very precise during this period and 210-Pb is at its limit.**

Response: Thank you very much for your suggestion. It is our negligence not to specify the number of dating numbers and resolution of paleoclimate records in detail, and we supplemented these parts in Table 3 in the revised version. As you suggested, paleoclimate records of Gonghai lake and Dali lake were added in the revised version. In this section, our aim is to reconstruct the regional moisture change by synthesizing the paleoclimate records to verify the continuous simulation. Therefore, we do not need to pay special attention to the dry and wet changes in the PI period, but focus on the matching degree of reconstructed results and simulated results throughout the Holocene. Both reconstructed moisture change and simulated water balance fluctuation exhibit a decreased trend since the early Holocene, giving our confidence that the simulations are useful for investigating the millennial evolution of the westerly winds and monsoons.

**5. In section 3.2, the authors state "Qinghai Lake, Hala Lake, Zhabuye Lake are typical lakes which are located in interactional transition zones between Asian monsoon and westerly winds, probably not following a single climate changing pattern". I would argue that many of the selected lakes in China, which they consider as being in the monsoon zones (see Fig. 6), were influenced both by the westerlies and the East Asian summer monsoon. Especially since the boundary of the monsoon was not static over time.**

Response: Indeed, due to various internal and external forces, the low-latitude monsoons and the mid-latitude westerly winds produce different intensities over time. The boundary of the East Asian summer monsoon will also be adjusted accordingly with the change of monsoon strength, leading to more complex and diverse evolution of Asian lakes. We modified this sentence in the revised version. The updated contents are reproduced below.

"Since the boundary of the monsoon will be adjusted accordingly with the change of East Asian summer monsoon strength, evolution of Asian lakes on the millennial scale probably not follows a single climate changing pattern (Wu et al., 2000; Editorial Committee of China's Physical Geography, 1984; An et al., 2012)."

**6. Structure of the manuscript Some parts of the result section belong to the discussion. While I understand that the authors must show that the lake simulations are valid and that, to do so, some interpretation is needed. I think that sections 3.3 and 3.4 should at the very least be moved to the discussion as they are focusing on the mechanisms driving the changes in water balance. Actually, I think that, given the nature of the data, this manuscript is a case where it would be beneficial to do a results and discussion section rather than separating them.**

Response: Thank you very much for your suggestion. Your suggestion provides a new perspective for discussing our study deeply. We combined the original result sections with the original discussion sections to create a "results and discussion" section with four parts.

**7. Terminology Several times in the manuscript, the authors refer to the Asian monsoon. To me it seems that what they call Asian monsoon is actually the East Asian monsoon. Especially since most of the selected records at the eastern edge of the simulated closed basins in Asia are roughly located at the northern limit of the East Asian summer monsoon (EASM). I think some precision is needed.**

Response: Thank you very much for your suggestion. We modified the "Asian monsoon" mentioned in this manuscript to the "East Asian summer monsoon" in the revised version.

**8. Discussion - Westerlies-monsoon interactions While studies have shown that trends in moisture changes in Westerly dominated arid Central Asia generally differ from those in EASM regions, owing to the fact that EASM rainfall does not reach this region, the opposite is not necessarily true. Records well into the region that the authors would consider as the East Asian monsoon region suggest an influence of the westerlies on moisture levels. The authors briefly discuss the interactions between the westerlies and the East Asian monsoon. However, I think the discussion would benefit from a more in-depth discussion of the relationship between the Westerly Jet and the EASM. For example, there are increasing evidence for a control of the Westerly Jet on the northward extent and timing of the EASM rainfall in East Asia (see for example: Chiang et al., 2015 QSR 108: 11-129; Herzschuh et al., 2019 Nat Comm 10; Nagashima et al., 2013 (Geochem Geophys Geosys 14: 5041-5053).**

Response: Thank you very much for your suggestion. The information you provided about the influence of the orientation and position of the westerly jet on the EASM rainfall give us a lot of help. Previous studies mostly focus on the complexity of climate change in the transition zone between the westerlies and Asian monsoon, and investigate the interplay of two global atmospheric circulation on the millennial scale. However, the impact of the seasonal progression of the westerly jet on the EASM rainfall has not been thoroughly discussed. We supplemented this issue in the section 3.4. The updated contents are reproduced below.

"The Northern Hemisphere westerlies shifting northward or southward has a significant impact on global atmosphere circulation and inevitably affects the monsoon systems. Quaternary ice sheets of the Northern Hemisphere in the LGM develop to its maximum extension, and consequent existence of persisting strong glacial anticyclone leads to the southward displacement of the westerly winds (Yu et al., 2000). Many researches suggested the Northern Hemisphere westerlies in the LGM move to the southwest of the United States and the eastern Mediterranean region (Lachniet et al., 2014; Rambeau, 2010). Therefore, the narrowed temperature difference between sea and land causes the East Asian summer monsoon weaken, and may further induces the strong westerly winds throughout the year and then the precipitation increases (Yu et al., 2000). Furthermore, a growing body of evidence shows that the position and orientation of the westerly jet (WJ) probably control

the Holocene East Asian summer rainfall patterns. A link between the northward seasonal progression of the WJ and the spatial pattern of East Asian summer monsoon precipitation shows that earlier northward progression of the WJ causes abundant precipitation at high-latitudes and less precipitation at low-latitudes (Nagashima et al., 2013). Especially the northward evolution of the WJ from south of the Tibetan Plateau and seasonal transition exert great influences on East Asian paleoclimate change (Chiang et al., 2015). Herzschuh et al. (2009) proposed that the position of summer monsoon rain band changes as the WJ axis shifts gradually southward, leading to the occurrence of spatiotemporal difference in Holocene China's maximum precipitation. In summary, the above views emphasize that the complex interaction between the monsoon and the westerly systems on the millennial scale should receive more attention."

**9. Speleothems The close similarity of the PCA1 time series with the speleothem records from Gongge and Hulu caves suggest it is a record of the East Asian summer monsoon. There is a long-standing debate about what the δ18O speleothem records from China represents. One view interprets the oxygen isotopic record from Chinese cave deposits as reflecting real rainfall changes and hence reflecting changes in the EASM. The other main view suggests that these the oxygen records (depending where they are located) reflect changes in the moisture source (Indian monsoon vs EASM) and that they do not directly represent changes in EASM. What can the present study contribute to that debate? I think it could be an interesting addition to this manuscript.**

Response: Thank you very much for your suggestion. We made corresponding supplement about the contribution of our results to the paleoclimate research of Chinese stalagmites in the revised version. The updated contents are reproduced below.

"The contribution rate of PCA1 and PCA2 is 51% and 14% respectively, therefore the following discussion mainly focuses on PCA1 with the high contribution rate. The PCA1 extracted from water balance simulation tends to represent the effective moisture fluctuation of closed basins in low-latitude monsoon regions, indicating a relative humid climate during the early-to-mid Holocene. By comprehensively analyzing a variety of paleoclimate proxies, Wang et al. (2017) suggested that moisture change revealed by the Australian monsoon, the East African monsoon and the Indian monsoon regions reaches the wettest status in the early Holocene, while the wettest condition in the East Asian summer monsoon regions occurs between 8 and 6 kyr. Likewise, Qin (1997) presented that the wettest period in the African and South Asian monsoon regions is the

early-to-mid Holocene, coinciding well with our results.

The climatic significance of the $\delta^{18}O$ in the Asian speleothem records is always a long-standing debate, and some influential hypotheses regard $\delta^{18}O$ of the monsoon regions as a proxy for "Asian monsoon intensity", "Indian monsoon intensity", "summer monsoon rainfall amount" and "circulation conditions" (Cheng et al., 2012; Chen et al., 2016). Although the climatic significance is controversial, it is well accepted that $\delta^{18}O$ changes should bear the imprint of variations in the oxygen isotopic composition of precipitation (Cheng et al., 2012; Chen et al., 2016). According to the close similarity of the PCA1 with the speleothem records from Dongge and Hulu caves, our simulations are more inclined to suggest that the $\delta^{18}O$ stalagmite records indicate the change in water vapor brought by the monsoons. In addition, we not only compared the PCA1 with the stalagmite records of Dongge Cave with controversial climatic significance, but also with the summer solar radiation at low-latitudes in the Northern Hemisphere. This comparison provides evidence for the view that the evolution of low-latitude monsoons is generally controlled by summer insolation in the Northern Hemisphere (Yuan et al., 2004; Chen et al., 2006; An et al., 2015). Thus, we further speculated that the water balance change in monsoon regions of global closed basins is mainly driven by mid-latitude and low-latitude summer solar radiation."

**Technical/minor comments Fig 3: Please provide letters to refer to each section of the figure both in the figure caption and the figure itself. I would also suggest putting both EOF figures on the left side and the PCA curves above the speleothem records. It would make the comparison of the curve easier.**

Response: Thank you very much for your suggestion. We modified the Figure 3 in the revised version. The updated contents are reproduced below.

[Figure]

"**Figure 3.** (a) Spatial distribution feature of EOF1, (b) PCA1 and PCA2 of simulated water balance change since the LGM, (c) Spatial distribution feature of EOF2, and (d) Comparison between stalagmite records and summer insolation: Stalagmite records come from Dykoski et al. (2005) and Wang et al. (2008), and summer insolation comes from Berger (1978)."

**Fig 5: please provide letters the refer to each time series, especially since the font size is quite small. If possible, increase the font size of the time series.**

Response: Thank you for pointing this out. We provided letters of each time series and increased the font size of the time series. The updated contents are reproduced below.

[Figure]

"**Figure 5.** Time series of (a) longwave radiation, (b) shortwave radiation, (c) temperature, (d) precipitation, (e) evaporation, (f) 500 hpa wind speed and (g) water balance change between 30°N and 60°N closed basins since the LGM."

**Section 3.3 and 3.4: EOF is not defined anywhere in the manuscript.**

Response: Thank you for pointing this out. We added section 2.3 to describe the mathematical methods. The updated contents are reproduced below.

"**2.3 Mathematical methods**

Linear tendency estimation is a common trend analysis method, which was chosen to measure the variation degree of simulated water balance in this paper. Besides, we also used the Empirical orthogonal function (EOF), a method of analyzing the structural features in matrix data and extracting the feature vector of main data, to examine spatially and temporally variability of simulated water balance. The spatial distribution of EOF first (second) mode is denoted by EOF1 (EOF2), and the time series of first (second) mode is denoted by PCA1 (PCA2)."

**Line 29: indicate rather than indicated. I would also remove monsoon after Australian and East African.**

Response: Thank you very much for your careful examination of the manuscript. We modified this.

**Line 30: Remove the And at the start of the sentence.**

Response: Thank you very much for your careful examination of the manuscript. We modified this.

**Lines 32-33. That sentence needs to be rephrased to something like ". . . the seasonal migration of the (ITCZ) profoundly influences the seasonality of the global monsoons."**

Response: Thank you very much for your suggestion. We rephrased this sentence in the revised version. The updated contents are reproduced below.

"Charney (1969) and Wang (2009) also proposed that the seasonal migration of the intertropical convergence zone (ITCZ) profoundly influences the seasonality of the global monsoons."

**Line 36: Please define LGM. This is the first time you mention it in the main body of the manuscript.**

Response: Thank you very much for your careful examination of the manuscript. We added the full name of LGM.

**Line 36: . . . southern regions of THE North American continent. . .**

Response: Thank you very much for your careful examination of the manuscript. We modified this.

**Line 51: Please define MH. This is the first time you mention it in the main body of the manuscript.**

Response: Thank you very much for your careful examination of the manuscript. We added the full name of MH.

**Line 51: remove one space between and and Pre-Industrial**

Response: Thank you very much for your careful examination of the manuscript. We modified this.

**Line 51: remove and at the start of the sentence.**

Response: Thank you very much for your suggestion. We modified this.

**Line 58: Capital letter for last**

Response: Thank you very much for your careful examination of the manuscript. We deleted this.

**Line 70: please define P-E. It is mentioned for the first time in the manuscript.**

Response: Thank you very much for your careful examination of the manuscript. We added the full name of P-E.

**Line 75: either remove And at the start of the sentence or combine with the previous one by for example writing: "(Peltier, 2004), while the vegetation. . ."**

Response: Thank you very much for your suggestion. We modified this. The updated contents are reproduced below.

"A remnant Laurentide ice sheet in the LGM and a modern-day ice sheet configuration in the MH and PI simulations are specified by the ICE-5G reconstruction (Peltier, 2004), while the vegetation is prescribed to modern values."

**Line 82: IN each grid cell not at**

Response: Thank you very much for your suggestion. We modified this.

**Line91: assumed rather than supposed**

Response: Thank you very much for your suggestion. We modified this.

**Lines 135-136: However, there are exceptions that lakes. . . Replace that by where**

Response: Thank you very much for your suggestion. We deleted this sentence.

**Lines 148-149: this sentence need to be rephrased, for example: " Comparing the simulations with the records, most simulations coincide with the upward. . ."**

Response: Thank you very much for your suggestion. We deleted this sentence.

**Line 135 "For better validating simulated results, reviewed and summarized the millennial-scale changing patterns in lake level of the closed basins since the LGM are particularly important."**

Response: Thank you very much for your careful examination of the manuscript. We deleted this sentence.

**Line 164: East Asian summer monsoon not East summer Asian. . .**

Response: Thank you very much for your careful examination of the manuscript. We modified this.

**Line 173: suggested change to: "According to. . . basins in the Northern Hemisphere, affected both by low-latitude monsoon and mid-latitude westerly winds, are ideal region. . ."**

Response: Thank you very much for your suggestion. We modified this. The updated contents are reproduced below.

"On the basis of the spatial characteristics of the EOF analysis, closed basins in the Northern Hemisphere, affected both by low-latitude monsoons and mid-latitude westerly winds, are ideal regions for revealing synergy of the westerly winds and monsoons."

**Line 179: "from A humid climate IN the early-mid Holocene to AN arid climate IN the late Holocene" not "from humid climate of the early-mid Holocene to arid climate of the late Holocene".**

Response: Thank you very much for your careful examination of the manuscript. We modified this. The updated contents are reproduced below.

"Simulated mean water balance curve corresponds well with mean moisture index in the Northern Hemisphere mid-latitude closed basins, indicating a transition from a humid climate in the early-to-mid Holocene to an arid climate in the late Holocene."

**Line 185: "in THE early-mid Holocene"**

Response: Thank you very much for your careful examination of the manuscript. We modified this.

**Lines 187-188: That sentence need to be rewritten.**

Response: Thank you very much for your suggestion. We reworded this sentence. The updated contents are reproduced below.

"Using paleoclimate modelling, Yu et al. (2000) mentioned that the low temperature during the glacial period causes a decrease of evaporation and a reduction of lake water loss, resulting in the appearance of high lake level."

**Lines 188-190: Do you still refer to Yu et al. (2000) there or to Fig. 5. This is not clear.**

Response: Thank you for pointing this out. Lines 188-190 describe Fig. 5 and we clarified it.

**Line 190: reaches A maximum not the**

Response: Thank you very much for your suggestion. We modified this.

**Line 221: experienced not experiences**

Response: Thank you very much for your suggestion. We modified this.

**Line 255-256: suggest edit: "Winter precipitations account for a large proportion of annual precipitations in these regions."**

Response: Thank you very much for your suggestion. We reworded this sentence. The updated contents are reproduced below.

[revised manuscript text omitted]

---

## Author Response (AR2)

Dear Editor,

Many thanks for your hard work and consideration on publication of our paper. We really appreciated all comments and suggestions very much. On behalf of my co-authors, we would like to express our great appreciation to you and reviewers. In this version, we revised the References format and the font of Figures according to the guidelines, as well as made some changes to the Acknowledgements. The marked-up manuscript is attached below.

Best regards,

Yours sincerely,

Yu Li

Corresponding author:

Name: Yu Li

Address: College of Earth and Environmental Sciences, Lanzhou University, Lanzhou 730000, China

E-mail: liyu@lzu.edu.cn

[revised manuscript text omitted]